# Biological Properties of Transition Metal Complexes with Metformin and Its Analogues

**DOI:** 10.3390/ph15040453

**Published:** 2022-04-06

**Authors:** Daniil A. Rusanov, Jiaying Zou, Maria V. Babak

**Affiliations:** 1Drug Discovery Lab, Department of Chemistry, City University of Hong Kong, 83 Tat Chee Avenue, Hong Kong SAR 999077, China; rd5411635841@gmail.com (D.A.R.); zcbejzo@ucl.ac.uk (J.Z.); 2Laboratory of Medicinal Chemistry, N. D. Zelinsky Institute of Organic Chemistry, Russian Academy of Sciences, Leninsky Avenue 47, 119991 Moscow, Russia; 3Department of Biochemical Engineering, University College London, Bernard Katz Building, Gower Street, London WC1E 6BT, UK

**Keywords:** metformin, phenformin, biguanides, metal complexes, transition metals, gold, lanthanides, prodrugs, antidiabetic, anticancer, antibacterial, antifungal, antimalarial

## Abstract

Metformin is a widely prescribed medication for the treatment and management of type 2 diabetes. It belongs to a class of biguanides, which are characterized by a wide range of diverse biological properties, including anticancer, antimicrobial, antimalarial, cardioprotective and other activities. It is known that biguanides serve as excellent N-donor bidentate ligands and readily form complexes with virtually all transition metals. Recent evidence suggests that the mechanism of action of metformin and its analogues is linked to their metal-binding properties. These findings prompted us to summarize the existing data on the synthetic strategies and biological properties of various metal complexes with metformin and its analogues. We demonstrated that coordination of biologically active biguanides to various metal centers often resulted in an improved pharmacological profile, including reduced drug resistance as well as a wider spectrum of activity. In addition, coordination to the redox-active metal centers, such as Au(III), allowed for various activatable strategies, leading to the selective activation of the prodrugs and reduced off-target toxicity.


**Table of Contents**
1. Introduction21.1. Brief Historical Outlook21.2. Diverse Therapeutic Applications of Metformin Derivatives31.3. Biological Consequences of Intracellular Interactions of Metformin with Endogenous Metals52. Biologically Active Metal Complexes with Metformin and Its Analogues72.1.Group III (Sc, Y and Lanthanides)72.2. Group IV (Ti, Zr, Hf)92.3. Group V (V, Nb, Ta)92.4.Group VI (Cr, Mo, W)132.5. Group VII (Mn, Tc, Re)152.6. Group VIII (Fe, Ru, Os)172.7. Group IX (Co, Rh, Ir)182.8. Group X (Ni, Pd, Pt)232.9. Group XI (Cu, Ag, Au)282.10. Group XII (Zn, Cd, Hg)342.11. The Role of the Metal Center in the Biological Activity and Potential Toxicity of Pre-Formed Metal-Metformin Complexes363. Conclusions and Future Outlook38Appendix A40References61

## 1. Introduction

### 1.1. Brief Historical Outlook

Based on the World Health Organization (WHO) list of essential medicines, metformin is considered an essential drug for people with diabetes [1]. Due to its safety profile and low cost, metformin has been used worldwide for the management of type 2 diabetes for more than half a century. In addition, metformin is also commonly used off label for the management of other medical conditions, such as polycystic ovary syndrome (PCOS) [2], insulin resistance and obesity [3]. Metformin belongs to the class of biguanides, which have a long medical history [4,5]. Long before the discovery of metformin, the extract from the *G. officinalis* plant was used by medieval European physicians to treat the symptoms that are now associated with type 2 diabetes. It was discovered that the most active extract was rich in guanidine (Figure 1). This chemical was synthetically produced at the end of the 19th century but was too toxic to be used in humans despite its hypoglycemic properties. In the 1920s, two synthetic biguanides—synthalin A and synthalin B—were introduced into clinical practice. Although their chemical structures consisted of two guanidine fragments separated by long aliphatic chains, these compounds were still somewhat toxic and were eventually replaced on the market by insulin. Despite marked structural similarity between guanidines and biguanides, the clinical potential of the latter was underappreciated until the discovery of an antimalarial drug saludrine (or proguanil), which was further modified to metformin hydrochloride (at that time called flumamine). It was reported that in 1949 flumamine was used in treating a local influenza outbreak in the Philippines [6]. Only in 1957 was the anti-diabetic potential of metformin rediscovered and the drug was marketed under the name glucophage (“glucose eater”) [7]. Subsequently, less polar analogues of metformin—phenformin and buformin—were reported to efficiently reduce blood glucose levels and were introduced to the market in some countries [4,5]. However, their use was associated with the incidence of lactic acidosis and, by the 1980s, they were eventually withdrawn from clinical use in most countries [4,5]. 

Unexpectedly, various retrospective epidemiologic analyses of patients with diabetes taking metformin or phenformin for prolonged periods of time revealed that these drugs reduced the incidence of cancer, as well as cardiovascular diseases [8,9,10]. In addition, some beneficial effects on liver and renal function were observed [8]. Overall, the antidiabetic and anticancer mechanisms of action of metformin are rather complex and have been described in detail elsewhere [9,11,12,13]. In brief, metformin and its analogues alter the energy metabolism of the cells, thereby acting as energy disruptors [14]. Metformin was shown to decrease the glucose absorption in the small intestine, increase glucose transport into cells and reduce plasma free fatty acid concentrations, thereby inhibiting gluconeogenesis [13,15]. In addition, metformin was shown to inhibit mitochondrial respiratory chain complex I and decrease hepatic energy status by activating the AMP-activated protein kinase (AMPK), which plays a central role in its mechanism of action [12]. The anticancer effects of metformin are exerted either directly or indirectly, i.e., via the induction of energetic crisis or systemic reduction of insulin levels [9,14]. Finally, the cardiovascular protective action of metformin might be related to its favorable actions on lipid metabolism, hypercoagulation, endothelial function, calcium signaling and platelet hyperactivity [16]. The promising epidemiological findings and extensive studies in various animal models prompted the re-evaluation of metformin, phenformin and their analogues for the use in other diseases [17,18,19]. Since the mechanisms of action, biomolecular targets, pharmacokinetics, pharmacodynamics and safety profiles of these antidiabetic drugs have already been established, some of the preclinical studies might be by-passed, leading to the accelerated approval of these drugs for the treatment of other diseases.

### 1.2. Diverse Therapeutic Applications of Metformin Derivatives

Biguanides are characterized by a diverse range of therapeutic activities, which have recently been summarized in the excellent review of Bharatam et al. [20]. Herein, we will briefly discuss the application of several metformin derivatives for the treatment and management of diseases other than diabetes, as well as touch upon several strategies for improving metformin activity. The interest in the development of biguanide compounds with antimalarial properties arose from the success of proguanil (paludrine, Figure 1), which has been frequently used since the 1940s. Even nowadays, chemoprophylaxis and treatment of malaria can be accomplished using malarone, which is a fixed-dose drug combination of proguanil and atovaquone [21]. Following the discovery of proguanil, global screening and synthetic efforts revealed several structurally similar compounds with antimalarial properties, including PS-15 (Figure 2). PS-15 and its analogues demonstrated excellent in vitro and in vivo activity against different resistant strains of *P. falciparum*, which causes the most dangerous form of malaria—falciparum malaria [22,23,24]. Subsequently, a large number of cyclic biguanides with antimalarial properties have been evaluated [20]. 

Moroxydine is a biguanide where one amine group has been replaced by the morpholine group. This compound efficiently inhibited both DNA and RNA viruses, including but not limited to herpes zoster virus, herpes simplex virus and adeno virus [20]. In addition, it was shown that moroxydine significantly reduced the duration of fever and pharyngitis [25]. As a result, it was extensively used in the 1960s for the treatment of viral infections such as influenza, measles and mumps. Although moroxydine hydrochloride is still used in several countries as an antiviral agent, its full biological potential has never been achieved. However, the temporary clinical success of moroxydine prompted the investigation of various compounds with a biguanide moiety, which revealed the prominent suppression of various DNA and RNA viruses, including HIV [20]. In light of the COVID-19 pandemic, moroxydine, metformin and other biguanides are considered for the treatment and management of SARS-CoV-2 [26,27,28]. 

The investigation of the antimicrobial properties of biguanides has led to the discovery of chlorhexidine and alexidine, as well as the polymeric compound polyhexanide (PHMB), which demonstrated strong bactericidal activity against a broad panel of gram-negative and gram-positive strains, as well as fungicidal activity, in particular against *C. albicans* and *streptococci* [20]. Chlorhexidine, alexidine and PHMB are widely used as disinfectants in human and veterinary practices, including surgeries, dental procedures and management of burns and mouth hygiene [20,29,30]. In addition, these drugs are used for the treatment of dermatological conditions, e.g. *Candida* infections [20,29]. Subsequently, synthetic efforts by medicinal chemists, as well as high-throughput screening of compound libraries, resulted in the discovery of novel biguanides with promising antimicrobial properties [20]. 

Following the epidemiological analysis of diabetic populations and the discovery of the correlations between the use of metformin or phenformin and a reduced risk of cancer incidence, both antidiabetic agents were investigated in various in vitro and in vivo cancer models [31,32]. Both compounds exhibited cytotoxicity in the millimolar or high micromolar concentration range and potentiated the anticancer activity of clinically used anticancer drugs, such as tamoxifen [33], doxorubicin [34], cisplatin [35] and other chemotherapeutic agents, both in vitro and in vivo. However, the potential use of metformin and phenformin in cancer treatment is hindered by serious drawbacks. According to the Biopharmaceutics Classification System (BCS) and Biopharmaceutics Drug Disposition Classification System (BDDCS), metformin is classified as a Class 3 compound (high solubility and low permeability). Due to its hydrophilic nature, metformin poorly penetrates through cellular membranes [36]; therefore, the desired anticancer activity can be achieved only at high doses. Phenformin is less polar than metformin; however, its anticancer effects in vitro and in vivo were also apparent only at high concentrations [37,38]. Since pathophysiological mechanisms underlying cancer may lead to lactic acidosis in most patients in different stages of the disease, chemotherapeutic regimens based on repeatedly high doses of metformin, or especially phenformin, would not be desirable. 

There are various strategies to overcome the difficulties associated with poor penetration of metformin and phenformin, including their encapsulation into nanocarriers [39], conjugation with targeting moieties [40], or development of prodrugs [41]. The simple modification of the metformin structure with pyrrolidine or furan heterocycles resulted in the formation of novel biguanide-based anticancer agents, HL156A [42,43] and NT1014 [44], respectively. Both compounds were characterized by increased AMPK activity and significantly enhanced cytotoxicity and in vivo activity in comparison with metformin; however, their cytotoxicity remained in the high micromolar range [44]. On the contrary, conjugation of the metformin backbone with a mitochondria-targeting triphenyl phosphine (TPP^+^) moiety via aliphatic chain linkers resulted in the formation of a series of compounds with markedly improved anticancer activity [40]. In particular, the lead compound mito-metformin (Mito-Met, Figure 2) was at least 1000 times more active than metformin against pancreatic ductal adenocarcinoma (IC_50_ = 1.1 μM and 1.3 mM for Mito-Met and metformin, respectively). It was shown that the anticancer mechanism of action of Mito-Met was based on AMPK activation as well as inhibition of mitochondrial respiration via inhibition of mitochondrial complex I and stimulation of superoxide and hydrogen peroxide formation [40,45]. 

One more approach to enhancing intracellular accumulation of metformin without inducing unwanted toxicity to healthy cells is the development of more lipophilic and pharmacologically inactive prodrugs, which would be biotransformed into metformin after absorption. In fact, the antimalarial compounds proguanil and PS-15 also serve as prodrugs since they transform into active cycloguanil metabolites inside the cells [46,47]. It was shown that proguanil and PS-15 activation were mediated by cytochrome P450 2C19 (CYP2C19) and cytochrome P450 3A4 (CYP3A4), respectively [48,49]. Besides malaria, metformin prodrugs might be useful for the treatment of various diseases, such as cancer [50], diabetes, Alzheimer’s disease [51] and others [52]. For example, metformin sulfenamide prodrugs demonstrated improved bioavailability and absorption (by ≈25%) and were readily converted into metformin upon interaction with intracellular thiols [41,53], thereby supporting the viability of the approach. These sulfenamide prodrugs exhibited beneficial effects on plasma haemostasis [52] and inhibited neurodegenerative acetylcholinesterase activity (AChE) [51].

### 1.3. Biological Consequences of Intracellular Interactions of Metformin with Endogenous Metals 

Biguanides serve as excellent *N*-donor bidentate ligands due to the presence of two imine groups in *cis*-positions and the localization of charge density on the terminal nitrogen atoms, which ultimately enhance the stability of the newly formed chelates. One of the first reports on the interactions of biguanides with transition metals, such as Cu or Pt, dates back more than a century ago [54]. Subsequently, a wide range of biguanides with various transition metals have been reported, and their molecular structures were supported by crystallographic evidence [55,56,57]. However, despite extensive structural and synthetic evidence, the biological role of metal-biguanide complex formation was not investigated until recently. It was found that in the absence of intracellular Cu, metformin-mediated AMPK activation in H4IIE liver cells was reduced by at least 50% [58]. The comparison of metformin, biguanide, propanediimidamide (PDI) and malonohydroxamamide (MHA) revealed that only those compounds that could form high-affinity pseudo-aromatic Cu complexes (metformin and biguanide, but not PDI and MHA) induced activation of AMPK signaling. In agreement, only biguanides, but not PDI, inhibited mitochondrial respiration and expression of gluconeogenic genes in H4IIE liver cells and suppressed hepatic glucose production in primary hepatocytes, suggesting that the antihyperglycemic properties of metformin might be Cu-dependent [58]. The computational analysis of Cu-binding energies revealed that the observed differences in biological effects exhibited by metformin, biguanide and PDI could not be explained by different Cu-binding energies [59]. Therefore, it was suggested that metformin and other biguanides might act as pH-sensitive Cu-binding prodrugs and their activation might occur at elevated mitochondrial pH levels, while PDI would require higher pH for the activation [59]. 

Since biguanides and other antidiabetic drugs are commonly characterized by their antimalarial properties, there might be some similarities between the therapeutic mechanisms of both diseases. Cysteine proteases play a role in both diseases and might be inhibited by endogenous metals. Therefore, it was hypothesized that biguanides might act as trans-compartmental metal shuttles and bring endogenous metals into the proximity of the active site of a cysteine protease with subsequent release of the metals upon dissociation [60]. It was shown that in the absence of the metals, biguanides did not appreciably inhibit falcepain-2 and cathepsin B activity, while in the presence of Zn(II) or Fe(III), both metformin and phenformin markedly increased the inhibitory effects of the metals by at least 25–55% [60]. The most prominent effects were observed by phenformin (0.02 μM) in the presence of an inactive concentration of Cu(II) (0.5 μM), which caused a remarkable 75% proteolytic inhibition [60]. It is possible that biguanides might play a similar metal-binding role in the context of diabetes, where they bind to the excess of Zn(II) ions on the surface of insulin, thereby preventing its degradation by cysteine proteases [60]. 

It was reported that not only the antidiabetic and antimalarial, but also the anticancer properties of metformin might be Cu-dependent. The concurrent treatment of several cancer cell lines with 400 μM of CuSO_4_ with increasing concentrations of metformin revealed a significant increase in metformin’s cytotoxicity [61]. However, it is not clear whether the observed effects were caused by Cu(II) alone or the combination of Cu(II) and metformin. It is well-known that excess of intracellular Cu levels results in the disturbance of cellular Cu homeostasis, oxidative stress and DNA damage [62,63,64,65].

In another work, the alkyne-containing metformin analogue was developed with the aim of establishing in situ labelling of metformin by means of click chemistry [66]. Although the analogue was characterized by higher cytotoxicity than metformin, it functionally phenocopied metformin in several in vitro models and therefore could be reliably used as a suitable metformin surrogate for the subsequent mechanistic investigations [66]. Based on the localization of the click-activated fluorescence, it was suggested that metformin surrogate was selectively accumulated in the mitochondria of breast cancer cells. Moreover, the intensity of the fluorescent signal significantly decreased upon co-incubation with metformin as a competitor. More detailed investigations confirmed the ability of biguanides to remove the redox-active Cu(I) ions from mitochondrial proteins and promote their oxidation to Cu(II), leading to an increase in mitochondrial Cu(II) ion levels and a decrease in mitochondrial Cu(I) ion levels [66], as predicted by the computational analysis [59]. Finally, to investigate whether the anticancer activity of metformin might be indeed linked to its Cu-binding ability, the effects of this drug on the epithelial-to-mesenchymal (EMT) transition were investigated. The EMT transition is commonly linked with the progression of cancer, the formation of metastases and increased tumor resistance. It is believed that Cu is an essential component of EMT; hence, it was hypothesized that the Cu-binding properties of metformin might lead to the suppression of EMT and decreased tumor stemness [66]. In agreement with the hypothesis, both metformin and its clickable analogue significantly reduced the expression of mesenchymal markers, such as fibronectin, vimentin, Zeb1, and decreased the proportion of CD24^−^/CD44^+^ cancer stem cells [66]. Interestingly, the anticancer activity of metformin and its mitochondria-targeting analogue Mito-Met was markedly enhanced in the presence of several Fe(III) chelators, such as deferasirox (DFX) [45]. Since metformin readily binds endogenous Zn(II), Cu(II) and Fe(III) and other metal ions and its cancer potency largely depends on Cu binding, it is plausible that DFX or other metal chelators might have reduced the competitive binding of metformin and other biguanides to Fe(III) and other metals. 

In the presence of endogenous metals, the biguanide moiety forms metal complexes in proportion to the relative binding affinities and metal availabilities of metals in cells and tissues. As a consequence, the simultaneous competitive binding with different metals might negatively affect the on-target biological activity of metformin and its analogues and induce off-target toxicity. A feasible approach is to administer pre-formed metal complexes of metformin and other biguanides, thereby delivering the most favorable biguanide/metal ratio for optimal biological function. Moreover, coordination of metformin to metal centers is expected to alter its uptake mechanisms and improve the intracellular accumulation and absorption in the bloodstream.

## 2. Biologically Active Metal Complexes with Metformin and Its Analogues 

In recent years, an increased interest in bioactive metal complexes has led to a multitude of studies describing the synthesis and biological activity of transition metal complexes with metformin and its analogues. In particular, metformin complexes with transition metals from Sc, Ti, V, Cr, Mn, Fe, Co, Ni, Cu and Zn families and lanthanides demonstrated antibacterial, antidiabetic, fungicidal and anticancer properties.

### 2.1. Group III (Sc, Y and Lanthanides) 

Although lanthanides are not considered biologically essential elements, they exhibit various biological properties, mainly due to their similarity to Ca [67]. The medicinal properties of lanthanides, including their antiemetic and antibacterial properties, were discovered two centuries ago. Since then various lanthanide salts, in particular Ce(III) and Ce(IV) compounds, demonstrated broad-spectrum antibacterial activity, which led to their clinical applications for burn management [67]. 

Coordination of 3 equiv. of metformin to Y(III), La(III), Ce(III) and Sm(III) nitrates resulted in the formation of complexes **1**–**4** with high coordination numbers in 60–70% yield (Figure 1) [68]. The antimicrobial activity of **1**–**4** against *S. aureus, B. subtilis, E. coli, P. aeruginosa* bacterial strains and *A. flavus* and *C. albicans* fungal strains was studied in comparison with metformin hydrochloride, the clinically used antibiotic tetracycline, and the antifungal agent amphotericin B using the filter paper disc method. As expected, all complexes revealed a broad spectrum of antimicrobial activities, while metformin was devoid of activity. Ce(III) complex **3** was the least active among all tested complexes, while **1**, **2** and **4** demonstrated similar activity to tetracycline in the majority of bacterial strains. The antifungal activity of **1** and **4** was comparable to that of amphotericin B in both fungal strains; however, **2** and **3** did not show any activity against *A. flavus*.

Nd(III) complexes **5** and **6** with metformin and its more lipophilic derivative were obtained from NdCl_3_⋅6H_2_O as a starting material (Figure 2) and their antidiabetic properties were tested in comparison with uncoordinated ligands and respective Nd(III) salt in Kunming white rats with induced diabetes [69]. It was shown that both complex **5**, Nd(III) salt and the respective ligand did not affect the blood sugar levels 2 h after the compounds were administered and only slight decrease in blood sugar levels was observed in rats treated with metformin and **6**. All compounds demonstrated similar, moderate antioxidant activity, which did not correlate with their antidiabetic properties [69].

Dy(III) complexes **7** and **8** with metformin derivatives were prepared from Dy(NO_3_)_3_⋅5H_2_O in ≈65% yield (Figure 3) and were also investigated in the context of diabetes [70]. The interactions of **7** and **8** with glucose were studied using spectrophotometric methods as well as viscosity measurements. It was revealed that **7** and **8** strongly bound glucose in aqueous solutions at physiological pH, which can be useful for the detection of glucose [70]; however, additional in vitro or in vivo experiments were not performed.

### 2.2. Group IV (Ti, Zr, Hf) 

To the best of our knowledge, there are only few examples of metformin complexes with group IV elements, and only one complex was described in the context of its biological activity. Coordination of metformin to a Zr(IV) center in the presence of 1,4-diacetylbenzene (DAB) resulted in the formation of complex **9** in excellent yield (Figure 4) [71]. The antibacterial and antifungal activities of **9** were tested against various bacterial and fungal cultures using the standard disk diffusion method in comparison with metformin, DAB and the antibacterial drug moxifloxacin.

When complex **9**, metformin and DAB were tested against two fungal strains, namely, *A. niger* and *C. albicans*, none of the compounds demonstrated fungicidal properties. However, complex **9** showed excellent antibacterial activity against all tested bacterial strains, namely, *E. faecalis*, *S. aureus*, *K. pneumoniae* and *Shigella*, which was 1.1–2.2 times lower than the activity of moxifloxacin. In contrast, both metformin and DAB did not show activity against any of the tested strains, indicating the important role of the Zr(IV) metal center in the antibacterial properties of complex **9**. It should be noted that various Zr(IV) complexes and nanoparticles showed marked antibacterial and antifungal activity, suggesting that the antibacterial activity of complex **9** originated from the metal center [72,73].

### 2.3. Group V (V, Nb, Ta) 

It is well-known that various V compounds are able to effectively normalize glucose levels both in vitro and in vivo, which makes them promising drug candidates for the treatment of diabetes [74,75]. Therefore, it was hypothesized that the combination of the antidiabetic drug metformin and the V center might lead to the synergistic activity of two fragments. Coordination of the two equivalents of metformin, phenformin or biguanide to an oxovanadium(IV) fragment resulted in the formation of complexes **10**–**12** of the type VO(L)_2_ in different yields (31–81%) (Figure 5) [76]. The investigation of antidiabetic activity of the oxovanadium(IV) metformin complex **10** was performed in Wistar diabetic rats in comparison with metformin and bis(maltolato)oxovanadium(IV) (BMOV), which previously demonstrated potent antidiabetic properties under similar experimental conditions. The diabetes was induced by a single intravenous injection of streptozotocin (STZ), resulting in blood glucose levels of over 13 mM. Subsequently, complex **10** was given to animals either via acute intraperitoneal (i.p.) injection at a dose of 0.12 mmol/kg or via acute oral gavage at a dose of 0.60 mmol/kg. The tail vein blood glucose levels were compared prior to drug administration and at selected times up to 72 h after drug administration. 

Following acute i.p. injection of complex **10,** BMOV and metformin, the response was observed only in rats treated with **10** and BMOV but not metformin. However, the glucose-lowering levels of complex **10** were less significant and persistent than the effects of BMOV, and an obvious side-effect in the form of diarrhea was observed. When the i.p. injection was replaced with acute oral gavage, only mild gastrointestinal effects were observed in all the treated groups. In total, 100% of rats responded to the treatment with complex 3 and their blood glucose levels returned to a normal range (less than 9 mM) within 24 h. However, the return of hyperglycemic levels after 72 h was observed for all rats. On the other hand, only 43% of BMOV-treated rats returned to hyperglycemic levels, indicating a more sustained response. No positive effects were observed in the metformin-treated group. These results indicated that oxovanadium(IV) metformin complex **10** was able to induce a significantly improved antidiabetic response in vivo than uncoordinated metformin, yet no synergistic or additive effects with metformin have been detected. 

Subsequently, the insulinotropic effects of complex **10** were investigated in comparison with [VO(pyrrolidine-N-dithiocarbamate)_2_] (VODTC) and VOSO_4_ using pancreatic islets isolated from rats with stimulated exocrine pancreatic secretion [77]. The islets were subsequently incubated with increasing concentrations (0.1–1 mM) of compounds of interest, followed by measurements of insulin concentrations. Among all the tested complexes, only VODTC induced significant insulin secretion, while complex **10** did not affect insulin release. 

Protein tyrosine phosphatases (PTPs) play an important role in the pathogenesis of various diseases, including diabetes and obesity. In an attempt to link the mild antidiabetic activity of complex **10** with its ability to inhibit PTPs, it was incubated with protein tyrosine phosphatase 1B (PTP1B), T cell protein tyrosine phosphatase (TCPTP), hematopoietic protein tyrosine phosphatase (HePTP) and Src homology 2 domain-containing tyrosine phosphatase 1 (SHP1), as well as alkaline phosphatase (ALP) [78]. Phenformin complex **11** and moroxydine complex **13** were used for comparison. As a result, all complexes demonstrated strong inhibition of PTP1B and TCPTP (IC_50_, 80–160 nM), slightly weaker inhibition of HePTP (IC_50_, 190–410 nM) and SHP-1 (IC_50_, 0.8–3.3 μM) and very weak inhibition of ALP (IC_50_, 17–35 μM). Complex **13** was twice less effective towards PTP1B, TCPTP and HePTP, than complexes **10** and **11**, while complex **11** demonstrated 3–4 times stronger inhibition of SHP-1 than complex **10 [78]**. The inhibition of PTP1B and ALP occurred via typical competitive inhibition of the active site of the enzymes. Based on these observations, it can be hypothesized that the structure of the biguanide to some extent might affect the selectivity of the complexes towards various PTPs and their antidiabetic properties in vivo.

To investigate whether the mode of metformin coordination to a V center might affect the antidiabetic properties of the resulting complexes, oxovanadium(IV) complexes with metformin-derived Schiff bases **14** and **15** were prepared (Figure 6) [79]. 

The diabetes in Swiss albino mice was induced by i.p. injections of alloxan (150 mg/kg/day). Subsequently, mice were treated via i.p. route with complexes **14**, **15** and uncoordinated metformin for 14 days (20 or 40 mg/kg). It was shown that metformin reduced blood glucose levels by 47–53% but did not show any effects on the total levels of serum cholesterol. In contrast, complexes **14** and **15** reduced blood glucose levels by up to 75% and decreased total cholesterol levels. None of the treatment regimens improved the integrity of pancreatic islets, which could possibly indicate that control of hyperglycemia was achieved by extrapancreatic mechanisms. 

Since oxovanadium(IV) metformin complex **10** did not demonstrate superior antidiabetic effects to the combination of metformin and vanadate fragment, another strategy has been employed. The decavanadate [V_10_O_28_]^6−^ consists of 10 octahedral vanadium centers and has various advantages over monomeric vanadates. In particular, it showed higher potency in lowering elevated blood glucose levels in diabetic rats. Considering the high anionic charge of decavanadate, its biological properties, in particular the ability to interact with biological membranes, are highly dependent on the counterions [80]. Since metformin affects hydrogen bonding in water, the replacement of the Na^+^ counterion in Na_6_[V_10_O_28_] with a metforminium cation resulted in a significant increase in solubility of the decavanadate salt in DMSO and the inhomogeneous environment of reverse micelles [81]. Subsequently, various metforminium decavanadates where metformin molecules served as counterions were prepared in moderate to good yields [81,82,83,84]. The effects of metforminium decavanadate **16** (MetfDeca, Figure 7), as well as uncoordinated metformin, were investigated in Wistar rats, which were given a hypercaloric (HC) diet for 3 months prior to treatment. Rats exposed to an HC diet were characterized by poor carbohydrate tolerance and the deposition of triglycerides in various organs, indicating insulin resistance. Metformin was given daily at a dose of 0.12M/kg together with the HC diet, and **16** was given twice a week at a dose of 2.5 μM/kg together with the HC diet for 30 days. Both treatments revealed significant improvement in morphometric regulation of body mass index (BMI) and fat percentage; however, only **16** demonstrated improvement in biochemical regulation. Importantly, the dose of **16** was 48,000 times lower than the dose of metformin, and the time of administration was reduced to twice a week, indicating the promising therapeutic potential of this compound. Additionally, the anti-diabetic effects of compound **16** were confirmed in other insulin-dependent and insulin-independent animal models [85]. 

Subsequently, the in vivo antidiabetic effects of **16** were simultaneously compared to metformin and NaVO_3_ [86]. Hyperglycemia and hypoinsulinemia were induced in Wistar rats via three days of i.p. applications of alloxan (150 mg/kg). Subsequently, rats were treated with either insulin (1 UI/100 mg/dL of glucose/day), metformin (350 mg/kg/day), **16** (3.5 μM/0.1 kg/day) or NaVO_3_ (3.5 μM/0.1 kg/day). It was shown that NaVO_3_ demonstrated improved hypoglycemic properties than metformin; however, the most pronounced hypoglycemic properties were demonstrated by insulin and **16**, reflected by restored redox balance in liver and muscles, as well as restored insulin levels. Importantly, this study revealed that complex **16** not only demonstrated improved anti-diabetic properties than metformin and monovanadate, but also mediated the regulation of hyperglycemia and oxidative stress through different pathways than monovanadate. 

Recently, it was reported that hypercaloric consumption in mice resulted in memory deterioration caused by impaired function of the hippocampus [87]. Therefore, it was investigated whether complex **16** could induce hippocampal regeneration and improve recognition memory in Wistar rats with metabolic syndrome [88]. Initially, rats were administered a normal or HC diet for 3 months and subsequently treated with **16** via oral gavage at a dose of 1.23 µg/0.1 kg twice a week for 60 days. As expected, complex **16** improved zoometric and biochemical parameters in rats given a HC diet. Importantly, **16** improved short-term recognition memory, diminished oxidative stress and improved antioxidant activity in rat brains. Administration of **16** reduced the inflammation of the hippocampus, characterized by reduced levels of pro-inflammatory cytokine TNF-α and increased levels of anti-inflammatory cytokine IL-10. In addition, **16** improved the morphology of hippocampal neurons, characterized by the rearrangement of dendritic trees and an increased number of dendritic spines in pyramidal neurons. Based on these observations, **16** might delay the onset of neurodegenerative diseases provoked by metabolic disorders.

Besides their role in the treatment of diabetes and other metabolic disorders, oxovanadium(IV) complexes with metformin and its structural analogues might be effective in the treatment of other diseases. For example, the ability of these complexes to irreversibly bind DNA might be useful for the treatment of cancer [89,90]. The incorporation of glycine or histidine into the oxovanadium(IV)-metformin backbone resulted in the formation of two water-soluble complexes, **17** and **18,** in excellent yields (Figure 8) [91]. 

The DNA-binding ability of these complexes was investigated using standard absorption titration experiments, fluorescence displacement experiments with EtBr, as well as viscosity measurements and gel electrophoresis, which suggested that complexes effectively bound DNA. Subsequent docking studies revealed that the strongest binding of **17** and **18** with DNA nucleotides occurred within the metformin binding pocket. Despite promising DNA-binding results, the anticancer activity of these complexes has not been investigated. 

Interestingly, the reaction of metformin with vanadyl sulfate resulted in the formation of a dinuclear oxovanadium(IV) metformin complex **19** ((VO)_2_(metf)_2_(SO_4_)_2_) with two SO_4_^2−^ anions acting as bridges (Figure 8) [92]. The activity of complex **19** at concentration 1 mg/mL against various gram-positive and gram-negative bacterial strains, as well as fungal strains was investigated using a standard disk diffusion method in comparison with uncoordinated metformin (1 mg/mL), streptomycin (10 mg/mL) and ketoconazole (10 mg/mL). Complex **19** demonstrated moderate activity against all tested bacterial and fungal strains, which was approximately 2–4 times lower than the activity of streptomycin and ketoconazole. However, these results cannot be directly compared due to significantly different drug concentrations. As expected, uncoordinated metformin was devoid of any significant activity against all tested bacterial and fungal strains. It was speculated that the improved antibacterial and antifungal activity of complex **19** in comparison with metformin might be related to the easier penetration of the metal complex through bacterial or fungal cell membrane; however, this hypothesis was not experimentally confirmed.

### 2.4. Group VI (Cr, Mo, W)

Similar to V complexes, Cr complexes with metformin demonstrate antibacterial, antifungal and antidiabetic properties. Cr(III) complex **20** with three bidentate metformin ligands were obtained by the reaction of CrCl_3_⋅6H_2_O with 3 equiv. of metformin in a 72% yield (Figure 9) [92]. Its activity was investigated against various bacterial and fungal strains under the same experimental conditions as complex **19**. In comparison with **19**, Cr(III) complex **20** demonstrated 1.5-, 2- and 1.8-fold stronger inhibition of *B. subtilis, P. aeruginosa* and A. niger strains, respectively, and 2.2-, and 1.6-fold weaker inhibition of E. coli and C. albicans strains, respectively. These results indicate that coordination of metformin to different metal centers allows for fine-tuning of the selectivity of the resulting complexes towards specific bacterial and fungal strains. 

While the antidiabetic properties of V compounds are well-documented, the role of Cr in diabetes is less established [93]. There is some evidence that Cr supplementation may improve the glycemic control in patients with diabetes [94]. Therefore, Cr(III) supplements are commonly used for diabetes and obesity treatment [95]. In addition, several Cr(III) complexes with various ligands induced sensitization of insulin signaling pathways in vitro and in vivo [96]. To investigate whether the combination of Cr(III) and metformin would result in enhanced antidiabetic properties, complex **20** (12.58 mg/kg and 25.16 mg/kg, corresponding to 1000 μg/kg and 2000 μg/kg of Cr) was administered orally to C57BL/6 mice with high-fat diet/STZ-induced diabetes in comparison with metformin (16.6 mg/kg) and CrCl_3_⋅6H_2_O (5.12 mg/kg, corresponding to 1000 μg/kg of Cr) for 30–60 days [97]. It was shown that all tested compounds efficiently lowered blood glucose and insulin levels by approximately 11–30%; however, complex **20** demonstrated the most pronounced effects on decreasing abnormal lipid levels. Importantly, both **20** and metformin did not cause any histopathological changes in the kidneys, pancreas, kidney and liver, indicating no sub-chronic toxicity. 

The most well-studied and best-selling Cr(III) supplement, which is believed to ameliorate insulin resistance and reduce the risk of cardiovascular diseases, is Cr picolinate [95]. Therefore, it was hypothesized that combination of Cr(III), dipicolinate and metformin might result in synergistic antidiabetic effects [98]. The X-ray diffraction analysis of complex **21** revealed that coordination sphere of a Cr(III) metal center was composed of two tridentate dipicolinate ligands, while metformin acted as counterion (Figure 9). The antidiabetic activity of **21** was assessed in mice with STZ-induced diabetes in comparison with CrCl_3_ and metformin. All tested compounds demonstrated only a moderate decrease of fasting blood glucose levels from ≈11.7 nmol/L to ≈7.8–8.6 nmol/L. However, complex **21** demonstrated significant reduction of total cholesterol and triglyceride levels, as well as partial normalization of high- and low-density lipoproteins. In agreement with initial hypothesis, the effects of **21** were more pronounced than the effects of metformin and respected inorganic Cr(III) salt. The post-mortem histological analysis of kidney and liver sections in treated mice did not reveal any pathological changes, indicating low toxicity of complex **21 [98]**.

In order to understand whether the replacement of the metal center from V to Cr might result in significant changes in antidiabetic activity, Cr(III) complex **22**, which is structurally similar to V complex **15**, has been prepared (Figure 10). The glucose-lowering properties of **22** were investigated in diabetic mice under the same experimental conditions as **15**. Additionally, the activity was compared to complex **23**, where biguanide fragment of metformin was not involved in the coordination to a metal center [99].

It was shown that both Cr complexes **22** and **23** decreased blood glucose levels in mice with alloxan-induced diabetes by up to 4.24 and 24.62%, respectively, at 20 mg/kg dose and up to 66–67% at 40 mg/kg. These results indicate that the coordination mode of metformin might play an important role in its antidiabetic effects. It should be noted that the structurally similar complex **15** demonstrated higher potency at a lower dose and equal potency at a higher dose. 

Additionally, a series of Cr(III) complexes **24**–**26** with metformin and other bidentate N-donor ligands has been prepared and their DNA-binding properties have been investigated (Figure 11) [100]. It was shown that these complexes could effectively bind DNA grooves, and the strength of DNA binding based on the DNA photocleavage study decreased in the following order: **26** > **25** > **24**. On the other hand, docking studies revealed that complex **25** and uncoordinated metformin were characterized by the highest docking scores. 

### 2.5. Group VII (Mn, Tc, Re)

There are several reports of Mn(II) complexes with various organic ligands that demonstrate some antibacterial and antifungal activity [101,102]. Mn(II)-metformin complexes were also investigated in the context of their antimicrobial activity. Coordination of 2 equiv. of metformin to a Mn(II) center resulted in the formation of octahedral complex **27** (Figure 12) [103]. This complex demonstrated a broad range of antibacterial activity against *E. coli*, *S. enteritidis*, *P. aeruginosa*, *B. subtilis, L. monocytogenes, S. aureus* and antifungal activity against *C. albicans*, which was 2–16-fold higher than the activity of metformin. However, no significant differences were observed between complex **27** and Mn(ClO_4_)_2_⋅6H_2_O salt, indicating the role of the Mn(II) center in the observed biological effects. 

The replacement of perchlorate axial ligands with acetate ligands in complex **28** did not result in significant changes in antibacterial or antifungal activity [104]. Additionally, preliminary anticancer activity of complex **28** has been tested against cervical carcinoma HeLa cells. While no significant cytotoxicity has been observed, **28** induced cancer cell cycle arrest at the G2/M phase. Surprisingly, other authors reported the antibacterial study of complex **29** with chlorido axial ligands, and no antibacterial activity has been observed [105]. We cannot unambiguously confirm the negative role of Cl axial ligands since the experiments were performed under different experimental conditions. 

^99m^Tc radiopharmaceuticals are widely used for diagnostic nuclear medicine due to the excellent nuclear properties of ^99m^Tc [106]. However, even though ^99m^Tc radionuclides are able to induce DNA double-strand breaks, their therapeutic use is hindered by their insufficient accumulation in cancer cells [107]. It was shown that conjugation of radionuclides to the DNA intercalator facilitated drug internalization and allowed for the ^99m^Tc decay in close proximity to DNA, leading to the formation of double-strand breaks [107]. Since metformin and its derivatives were shown to effectively bind minor/major groove of DNA in both intercalative and non-intercalative mode [108,109], they might enhance the accumulation of ^99m^Tc radionuclides in the vicinity of DNA. Tricarbonyl ^99m^Tc(I) complex **30** with phenformin was prepared in two steps starting from readily available Na^99m^TcO_4_ (Figure 13) [110]. Complex **30** demonstrated high stability in the presence of histidine and cysteine and moderate stability in rat serum and might exhibit some potential as a radiotherapeutic agent; however, its interaction with DNA has not been studied [110]. 

### 2.6. Group VIII (Fe, Ru, Os) 

It was reported that various Fe complexes demonstrated a broad range of anticancer and antibacterial activities [111,112,113]. For example, Fe(III) complexes with Schiff base-derived ligands significantly inhibited the growth of gram-positive bacteria, possibly through the induction of ferroptosis [112]. Structurally different Fe(III)-metformin complexes **31**–**33** also demonstrated some antibacterial activity [105,114]. It was shown that the product of the reaction between metformin and FeCl_3_⋅6H_2_O was dependent on the amount of added base (Figure 14) [114]. In particular, the addition of 1 equiv. of KOH (based on metformin) resulted in the formation of dinuclear bridge complex **31**, while the addition of 0.5 equiv. of KOH yielded a typical square planar coordination complex of the type ML_2_. Subsequently, the antibacterial activity of both complexes and metformin has been tested against *S. aureus, P. aeruginosa, E. coli, K. pneumoniae* and the fungal strain *C. albicans* using the disk diffusion method. As expected, uncoordinated metformin did not show any activity, except for *S. aureus* and *E. coli*, and its coordination to Fe(III) resulted in a significant improvement in antibacterial and antifungal properties. The structure of the complexes determined the selectivity towards the following particular strains: while complex **31** was more active towards *P. aeruginosa* and *E. coli*, complex **32** was more selective towards *S. aureus and K. pneumoniae.* On the contrary to Mn complex **29**, structurally similar Fe(III) complex **33** demonstrated some inhibitory potential towards *E. coli, P. aeruginosa* and *S. aureus* [105].

Ru(II) and Ru(III) complexes with biological properties have gained considerable popularity in recent decades [115,116,117]. The initial interest in Ru anticancer complexes was centered on the belief that Ru can mimic Fe and can be selectively transported to cancer cells with high Fe demand by Fe transporters. Nowadays, the role of transferrin in the transport of Ru-based drug candidates is debatable, and the exact mechanism of their subcellular localization remains elusive [118,119]. Nevertheless, the success of *trans*-[tetrachloridobis(1H-indazole)ruthenate(III)] (KP1019) or its sodium salt (KP1339 or IT-139 or BOLD-100) in clinical trials (e.g., NCT04421820, NCT01415297) [120,121], suggests that development of Ru-based anticancer complexes is a viable therapeutic strategy. In particular, half-sandwich Ru(II) anticancer complexes are interesting from the perspective of their easy functionalization and conjugation with various biologically-active fragments [116]. Typically, DNA is not considered as the main biomolecular target of half-sandwich Ru(II) complexes, since the large number of Ru(II) complexes demonstrated a strong preference towards thiol-containing blood serum proteins, such as bovine serum albumin (BSA) [122]. Therefore, it was hypothesized that coordination of metformin, which was shown to effectively bind minor/major groove of DNA [108,109], might enhance the interactions of half-sandwich Ru(II) complexes with DNA, leading to the DNA damage [123]. Complexes **34** and **35** with metformin were prepared in 74–86% yields using standard (η^6^-*p*-cymene) or (η^6^-benzene)Ru dimers as starting materials (Figure 15). These drug candidates were active against human breast carcinoma MDA-MB-231 cells, human lung carcinoma A549 cells, as well as human ovarian carcinoma A2780 cells in the range of ≈ 8–30 µM, while metformin was not cytotoxic. On average, **34** was at least 1.5-fold more active than **35** in all cancer cell lines. Importantly, **34** and **35** were not toxic against healthy embryonic kidney HEK293 cells, thereby providing a wide therapeutic window for anticipated treatment strategies. Based on competitive fluorescence assays and docking simulations, it was concluded that **34** and **35** bound to DNA in a non-intercalative manner. The propensity of metformin for strong hydrogen bonding with DNA nucleobases [108,109] significantly contributed towards the DNA-binding affinity of the complexes [123]. In addition, viscosity measurements and gel electrophoresis studies with the supercoiled pUC19 DNA plasmid revealed covalent adduct formation with DNA. As expected, some binding interactions with BSA were observed, which were more pronounced for complex **35** than for **34** [123]. We hypothesize that complexes **34** and **35** might be transported into cancer cells using BSA as a carrier, where they subsequently induce extensive DNA damage, leading to apoptosis. 

Since novel compounds were not toxic to normal cells, their antidiabetic properties were investigated by measuring α-amylase activity, which typically prevents the absorption of glucose in diabetic patients. It was shown that both complexes could effectively inhibit α-amylase activity at a high micromolar range; however, they were at least twice less efficient than the standard drug acarbose [123]. 

### 2.7. Group IX (Co, Rh, Ir)

The group of Co is widely presented by the whole range of structurally different Co(II) metformin complexes with various biological properties, including antibacterial, antifungal, antiviral, anticancer and antidiabetic complexes. The reaction of metformin with CoCl_2_⋅6H_2_O in a 1:1 ratio resulted in the formation of tetrahedral complex **36** (Figure 16) [124]. It was determined by the liquid medium dilution method that the antibacterial activity of complex **36** against *E. coli*, *K. pneumoniae* and *P. aeruginosa* was lower than metformin’s activity; however, this complex demonstrated good inhibitory potential towards *B. subtilis* (MIC 64 μg/mL) and *S. aureus* (MIC 128 μg/mL).

When metformin was added to a Co(II) center in the presence of additional chelating and non-chelating ligands, such as water, DAB or Schiff-bases, resulting complexes adopted octahedral geometry (Figure 16) [71,125]. On the contrary to **36**, complex **37** was devoid of activity against various bacterial and fungal strains, including *S. Aureus* and *K. Pneumoniae* [71]. Surprisingly, the activity of **37** against gram-negative *Shigella* bacteria was even higher than the activity of the antibacterial drug moxifloxacin [71]. It is known that Schiff-bases are commonly characterized by the wide range of biological properties, including antimicrobial activity [126]; therefore, the combination of metformin, a Schiff-base and Co(II) center was expected to demonstrate an improved antibacterial profile [125]. As a result, complex **38** demonstrated slightly improved activity towards *E. coli* (zone of inhibition: 11.29 mL (**38**), 10.41 mm (metformin) and 7.14 mm (Schiff base). However, no improvement in activity against *B. megaterium* has been observed (zone of inhibition: 8.29 mm (**38**), 10.07 mm (metformin) and 8.01 mm (Schiff base)).

In one of the most recent studies, a series of Co(III) **39**–**43** complexes with metformin and its analogues via three-step synthesis (Figure 17) [127,128]. In the first step, biguanide ligands were coordinated to a Co(II) salt in an alkaline medium. In the second step, the resulting Co(II) complex was oxidized to a Co(III) complex using H_2_O_2_ and in the last step the OH^-^ counterion was replaced by Cl^-^ using diluted HCl. It should be noted that complex **44** was not converted to chloride and the moroxydine ligand was coordinated in a deprotonated form. Subsequently, the antiviral activity of novel complexes was tested against the influenza virus in comparison with [Co(En)_3_]Cl_3_, where En = ethylenediamine. Madin-Darby canine kidney (MDCK) cells were infected with the A/California/07/09 (H1N1pdm09) influenza virus and, 30 min after infection, the cells were incubated with compounds of interest for 72 h. The inhibition of viral replication was detected by the neutral red uptake assay or microscopy. While complexes **39, 41, 42** and **44** did not show any viral inhibitory potential, **40** and **43** demonstrated significant inhibition of influenza virus replication in 125–250 μg/mL dose range. However, at 250 μg/mL, complex **40** was highly toxic to the mammalian cells, while **43** demonstrated an excellent selectivity index (at least 8 times more selective towards viral cells) [127]. Co(En)_3_Cl_3_ did not show any inhibitory potential, indicating the role of biguanide ligands [127]. In another work, the cytotoxicity of **39** was tested against mouse muscle C2C12 cells and human liver carcinoma HepG2 cells [129]. Similar to **43**, **39** did not show significant toxicity, indicating that it can be safely used as an antiviral agent or for other purposes [129]. Surprisingly, despite the lack of activity of complexes **41**, **42** and **44** against influenza virus, they demonstrated excellent inhibitory potential of herpes simplex virus type 2 strain MS (HSV-2) [128]. In particular, complex **41** inhibited HSV-2 at ED_50_ = 6.25 μg/mL and was at least 16 times more selective towards the virus than towards mammalian cells [128]. These results revealed the excellent therapeutic potential of Co-biguanide complexes as antiviral agents and the drastic influence of the biguanide ligands on the antiviral activity and selectivity of the complexes.

The biological properties of Co complexes have been extensively investigated for the last 70 years, and the anticancer potential of Co is well-documented [130]. Since Co is an essential trace element, which is particularly required for the biosynthesis of vitamin B12, the disruption of Co homeostasis can be used as an effective therapeutic strategy in cancer. In addition, the fine-tuneable redox-activity of Co complexes allows for easy delivery of bioactive ligands to cancer cells. The anticancer potential of several Co complexes with metformin has also been investigated. 

Co(II) complexes **45**–**48** (Figure 18) with metformin and bidentate *N*-donor ligands demonstrated the ability to bind DNA within the binding pocket of metformin, similar to Cr complexes **24**–**26** (Figure 11) [131,132]. It should be noted that both the Cr and Co complexes with metformin and *o*-phenylenediamine were characterized by the highest DNA docking scores. In addition, the antidiabetic activity of complex **48** was investigated in mice with STZ-induced diabetes. This complex significantly decreased blood glucose levels as well as normalized lipid profiles; however, no improvement in comparison with metformin has been observed [132]. The anticancer activity of Co(II) complex **49** with two metformin ligands and two nitrate anions (Figure 18) contributing to the octahedral coordination sphere has been investigated in vitro against Ehrlich ascites carcinoma (EACC) cells. As expected, metformin was devoid of cytotoxicity, while incubation of cancer cells with 300 μg/mL of **49** resulted in only 19% of residual cell viability [133]. Since both metformin and Co complexes were reported to act as antioxidants [134,135], the antioxidant activity of **49** was tested in comparison with uncoordinated metformin using a stable free radical, α,α-diphenyl-β-picrylhydrazyl (DPPH). Both **49** and metformin demonstrated relatively high antioxidant activity of 62 and 41%, respectively [133]. 

Ir(III) complexes represent a promising class of metal-based biologically active compounds due to the relative inertness of the low-spin 5d electronic configuration of the outer shell of Ir(III) and the relatively high stability of its complexes [136]. Sadler et al. prepared a comprehensive series of half-sandwich Ir(III) complexes with metformin and its analogues, aiming to investigate whether the antimicrobial properties of the complexes can be fine-tuned by the choice of substituents on π-bonded arene or biguanide ligands (Figure 19) [137].

Subsequently, the antibacterial activity of the resulting 16 and 18-electron complexes in comparison with several uncoordinated biguanide ligands was determined against a panel of gram-positive and gram-negative bacterial strains, as well as fungal strains. Importantly, some relationships between the structure, hydrophobicity and antimicrobial activity of the complexes have been established. All tested ligands, including metformin, as well as more hydrophilic complexes **50** and **51** with metformin, were devoid of activity against various pathogenic bacterial and fungal strains with minimum inhibitory concentrations (MIC > 32 μg/mL). On the other hand, more lipophilic complex **52** with metformin demonstrated increased activity against gram-positive strains, probably due to the higher level of penetration through the bacterial membrane. Other lipophilic complexes with phenyl and biphenyl substituents **53**–**58** demonstrated excellent activity against gram-positive (MIC 0.125–1 μg/mL) and gram-negative bacterial strains (MIC 1–16 μg/mL), with the exception of *P. aeruginosa*, which is known to have poor membrane permeability. Interestingly, complexes **59**–**62** with a sulfonyl group with aromatic substituents demonstrated similarly high activity against gram-positive strains and MRSA and no activity against gram-negative strains. All lipophilic complexes, with the exception of **50** and **51,** demonstrated significant activity against the fungal strains *C. albicans* and *C. neoformans*. With regards to the effects of halido ligand X, no clear structure-activity relationships between **55**, **57** and **58** were observed. Importantly, novel Ir complexes demonstrated high levels of selectivity towards microbial organisms vs. mammalian cells, in particular complex **56** (selectivity factor (SF) values range between 8 and >256). Importantly, the antimicrobial activity of Ir(III) complexes was linked with the specific mechanism of action. It was shown that ROS generation, DNA binding or cell wall targeting were responsible for the observed antimicrobial effects. On the other hand, reaction with intracellular thiols, such as L-cysteine, resulted in the rapid release of biguanide ligands and (arene)Ir(cysteine) species, possibly leading to the inhibition of protein biosynthesis. Overall, Ir(III) complexes might selectively deliver metformin and analogous biguanide species to the cells, which otherwise could not penetrate the microbial membrane. This example represents the importance of metal coordination of metformin and its analogues, leading to improved penetration, novel mechanisms of action and biomolecular targets. 

Recently, Mao et al. prepared heteroleptic Ir(III) complexes with metformin **63**–**65** in moderate to good yields, starting from chloro-bridged cyclometalated Ir(III) dimers with subsequent counterion exchange (Figure 20) [138]. Novel complexes were tested against a panel of cancer cell lines in normoxic and hypoxic conditions in comparison with clinically used anticancer drug cisplatin and a structurally similar Ir(III) complex without biguanide ligand. In general, complexes **63**–**65** were significantly more cytotoxic than cisplatin, in particular in hypoxic conditions. In both normoxic and hypoxic conditions, the cytotoxicity decreased according to the following trend: **63** > **65** > **64**. On the other hand, the Ir(III) complex without metformin was characterized by decreased anticancer activity in hypoxic conditions, indicating the role of the biguanide ligand. These differences were corroborated by the ability of **63**, but not analogous Ir(III) complex to reduce the expression of hypoxia inducible factor-1α (HIF-1α). The mechanism of action of **63** was linked to the ROS generation and interference with mitochondrial respiration of cancer cells. In addition, complex **63** demonstrated promising anti-invasive and anti-inflammatory potential. Similar to a previously described study, complexes **63**–**65** readily reacted with glutathione (GSH), resulting in the displacement of the metformin ligand. Therefore, the observed effects might be attributed to the selective release of metformin into the intracellular cancer environment. 

### 2.8. Group X (Ni, Pd, Pt)

The majority of metformin complexes with group 10 transition metals have been investigated in the context of their antibacterial activity. In particular, various Ni(II) complexes demonstrated significant activity against a panel of bacterial strains [139,140,141]. The reaction of 2 equiv. of metformin with various Ni(II) salts (Figure 21) resulted in the formation of complexes **66**–**69**. In contrast to the structurally similar Mn complexes **27** and **28**, Ni complexes **66**–**68** were obtained as square planar tetracoordinate complexes with perchlorate, acetate or chloride anions outside of the coordination sphere [103,104]. On the other hand, complex **69,** which was obtained by the reaction of metformin and NiCl_2_⋅6H_2_O in water, was characterized by the hexacoordinate octahedral coordination sphere. It should be noted that the structure of **69** was not confirmed by X-ray diffraction. 

Both **66** and **67** demonstrated some inhibitory activity against the panel of bacterial strains, including *E. coli, P. aeruginosa* and *S. enteritidis,* with MIC of between 256–512 μg/mL, while corresponding inorganic Ni(II) salts were devoid of activity. In general, tetracoordinate Ni-metformin complexes were less effective than structurally similar Mn complexes; however, **67** demonstrated exceptionally high activity against the *L. monocytogenes* strain with an MIC = 4 μg/mL. Furthermore, hexacoordinate complex **69** was more effective than the structurally similar Mn complex **29 [105]**. It seems that coordination of chlorido ligand to the Ni(II) center did not significantly affect the activity of **69** in comparison with **68,** and both complexes were characterized by excellent inhibitory potential against several gram-positive and gram-negative bacterial strains [142]; however, direct comparison cannot be performed due to the differences in experimental conditions. The reaction of NiCl_2_⋅6H_2_O with metformin in the presence of other ligands, such as DAB [71] or a tridentate chelating ligand iminodiacetic acid [142] yielded penta- and hexacoordinate complexes **70** and **71** (Figure 22). In contrast to structurally similar Zr complex **9**, **71** was not active against all tested bacterial and fungal strains except *K. pneumoniae*, while complex **70** demonstrated broad antibacterial activity, comparable to **68**. It should be noted that uncoordinated iminodiacetic acid and metformin ligands also demonstrated some antibacterial activity under the same experimental conditions; however, the activity was lower. 

Since complexes **68** and **70** demonstrated promising antimicrobial activity, their anticancer activity against liver cancer HepG2 cells has been investigated in comparison with metformin and iminodiacetic acid [142]. All compounds demonstrated marginal cytotoxicity in the mM range, yet the activity of **68** and **70** was at least 2–4 times higher than the activity of the ligands, indicating the importance of the Ni(II) center. The observed cytotoxicity might be related to the ability of complexes **68** and **70** to irreversibly bind blood proteins such as albumin. 

Metformin can be coordinated with a metal center as a part of a macrocycle. For example, macrocyclic Ni complexes **74** and **75** were obtained in two steps via the intermediate formation of a square planar complex **73** with two deprotonated metformin ligands (Figure 23) [143]. Despite relative structural similarities, **73**–**75** demonstrated differential selectivity towards various bacterial and fungal strains. While complexes **73** and **74** were equally active against *S. aureus*, *E. faecalis*, *E. faecium*, *E. coli*, *P. aeruginosa*, *C. albicans* and *C. parapsilosis* (MIC values ≈100–300 μg/mL), complex **75** demonstrated selectivity towards *C. albicans* and *C. parapsilosis* (MIC values <100 μg/mL). Importantly, these compounds inhibited bacterial biofilm formation, which is commonly associated with nosocomial infections. Similar to **68** and **70**, Ni complexes **73**–**75** induced relatively marginal anticancer effects in human ileocecal adenocarcinoma (HCT8) and cervical cancer (HeLa) lines, as reflected by insignificant induction of apoptosis and cell cycle interference. In agreement, the structurally similar complex **72** was devoid of cytotoxicity against mouse muscle C2C12 cells and human liver carcinoma HepG2 cells [129]. Subsequently, the drug-likeness of **73**–**75** was assessed by various computational methods using pharmacokinetic bioinformatic databases. Complexes **73** and **74** presented good drug-like features, but only **74** displayed reasonable intestinal absorption and suitable blood-brain-barrier (but not central nervous system) permeability. Based on the computational predictions, all complexes were not toxic to the liver; however, **73** could cause skin sensitization. In addition, complexes with macrocyclic ligands were predicted to inhibit protease activity. 

The DNA-binding activity of a series of heteroleptic octahedral Ni complexes with metformin and En or other bidentate *N*-donor ligands **76**–**80** (Figure 24) has been investigated using various spectrochemical methods [132,144]. As expected, all complexes were able to bind DNA grooves, similar to Co complexes **45**–**48** and Cr complexes **24**–**26**, suggesting that octahedral complexes with metformin and other bidentate *N*-donor ligands demonstrate similar DNA binding properties, independent of the metal center. In addition, complex **80** demonstrated some anti-diabetic properties, similar to Co complex **48**. 

Pd(II)-metformin complexes **81**–**85** (Figure 3) were prepared using PdCl_2_ or Pd(OAc)_2_ as starting materials under the same experimental conditions as analogous Ni(II) complexes and their antimicrobial activity was compared [71,105,145,146]. Pd(II) complex **81** demonstrated similar antibacterial activity as Ni(II) complex **68**. However, in contrast to **68**, **81** strongly inhibited *A. flavus* and *C. albicans* fungal strains [105]. Similarly, macrocyclic complexes **83** and **84** showed significantly higher antimicrobial activity (MIC values ≈16–62 μg/mL) than structurally analogous Ni(II) complexes **74** and **75 [145]**. In addition, **83** and **84** effectively induced apoptosis and necrosis in HeLa cells [145], while **74** and **75** were virtually inactive [109]. Complex **85** was equally active against *E. faecalis* and *Shigella* as Zr complex **9**, but did not display any activity against *K. pneumoniae* and *S. aureus* [105].

It is known that cyclometalated complexes of Pd(II) and Pt(II) are often characterized by excellent anticancer activity [147,148,149]. A series of cyclopalladated metformin complexes with various substituents on the benzylamine moiety have been prepared according to the synthetic route described in Figure 25. The anticancer activity of complex **86** has been tested against HeLa, MCF7 and A549 cancer cell lines in comparison with complex **82**, uncoordinated metformin and the clinically used anticancer drug cisplatin [146]. With the exception of A549, **86** was 2–5-fold more cytotoxic than **82**, suggesting a beneficial role of cyclometalated fragments. Both complexes displayed cytotoxicity in the high micromolar range and were less active than cisplatin but significantly more active than metformin, which is known to display cytotoxicity in the high millimolar concentration range. The anticancer activity of **82** and **86** was linked with their DNA intercalation properties, which were confirmed by UV-vis and fluorescent spectroscopy. The methylene blue displacement assay suggested that DNA intercalation occurred via the metformin moiety. In addition, **82**, **86** and metformin were shown to effectively interact with BSA; however, the competition experiments revealed the differences in the binding sites between complexes **82** and **86** and metformin [146]. 

Since the discovery of cisplatin, Pt(II) complexes have been extensively investigated for their anticancer properties. In general, these complexes exhibit their anticancer activity as a result of DNA binding, which leads to the damage of healthy cells and severe side-effects [150,151]. Pt(IV) complexes are typically less toxic since they can be selectively activated in cancer cells by various triggers [152,153]. The first synthesis of Pt(II)-metformin complex **87** from *cis*-dichlorobis(dimethyl sulfoxide)platinum(II) and metformin hydrochloride was performed in 1995 (Figure 26) [154]; however, no biological properties of this compound were investigated. Subsequently, Pt(IV) complex **88** was prepared from K_2_PtCl_6_ in a 27% yield and its anticancer properties were studied on cisplatin-sensitive and cisplatin-resistant P388 leukemia cells in comparison with cisplatin [155]. 

Both **90** and cisplatin demonstrated excellent cytotoxicity in low micromolar concentration ranges in cisplatin-sensitive cells (IC_50 (48 h)_ = 0.86 ± 0.08 and 1.22 ± 0.30 μM for **90** and cisplatin, respectively) and 5–12-fold lower cytotoxicity in cisplatin-resistant cells (IC_50 (48 h)_ = 4.38 ± 0.53 and 14.30 ± 0.95 μM for **90** and cisplatin, respectively) [155]. Even though it was shown that both compounds caused similar cycle perturbations at equimolar concentrations, namely, equal levels of cellular accumulation at G2/M phase, the lower resistance factor for **90** indicates the differences in its mechanism of action in comparison with cisplatin. Inspired by the promising in vitro results, the in vivo effects of **90** (6.25–50 mg/kg, *i.p.* route) were investigated in B6D2F1 mice with P388 xenografts in comparison with cisplatin [155]. Complex **90** demonstrated significant improvement of mouse life span (an increase of 59%) at a maximum tolerated dose of 25 mg/kg, while cisplatin demonstrated a marked 192% improvement at 10 mg/kg. The marked differences between in vitro and in vivo results suggest possible differences in the pharmacokinetic behavior of these compounds. 

Several Pt(II) and Pt(IV)complexes were investigated as potential antimicrobial agents. Structurally similar Pt(II) and Pd(II) complexes were prepared from the corresponding salts (Figure 26) [124]. In general, both complexes did not show any prominent activity against a panel of bacterial strains (MIC 512–1024 μg/mL); however, drastic differences were observed in *B. subtilis* and *S. aureus* [124]. Pt(II) complex **88**, as well as uncoordinated metformin, were devoid of activity against *S. aureus* Pd(II) complex **89** was moderately active (MIC 256 μg/mL). On the contrary, **89** was relatively inactive against *B. subtilis*, while **88** demonstrated strong inhibitory potential (MIC 64 μg/mL), which was 2-fold higher than the activity of metformin and similar to the activity of Co complex **36**. In addition, the antimicrobial activity of Pt(IV) complex **91** with four monocoordinated deprotonated metformin ligands has been investigated using the disk diffusion method (Figure 26) [105]. Additionally, **91** was moderately active against all tested bacterial and fungal strains, and its inhibitory potential was comparable to that of Pd(II) complex **81**. 

### 2.9. Group XI (Cu, Ag, Au)

The antibacterial properties of Cu have been known since ancient civilizations [156]. Cu surfaces and materials were shown to effectively inhibit bacterial biofilms, including methicillin-resistant *S. aureus*, resulting in a significant reduction in hospital-acquired infections [157,158]. Aiming to understand, whether Cu(II)-metformin complexes might have a therapeutic potential as antimicrobial agents, a large panel of Cu complexes has been tested against various bacterial and fungal strains and compared with structurally similar complexes with other metal centers (Figure 27 and Figure 4). Complex **92** was prepared by the condensation of metformin and readily available 2-pyridinecarbaldehyde in the presence of Cu(ClO_4_)_2_⋅6H_2_O in a 76% yield (Figure 27). Subsequent nucleophilic addition of methanol resulted in the formation of **93** with a 26% yield, whose structure was confirmed by X-ray diffraction [159].

The antibacterial activity of **92**, **93** and metformin was tested against *S. aureus, B. pumilus, Salmonella* and *E. coli* in the range of 1.25–10 mmol/L using the agar diffusion method. All compounds caused inhibition of bacterial growth, as reflected by the diameter values of the inhibition zone of around 11.2–21.6 mm. The inhibitory potential of metformin was not strongly dependent on the dose, while complexes **92** and **93** demonstrated up to a 1.4-fold increase in the inhibition zone diameter at higher concentrations. Metformin **92** and **93** were equally moderately potent against E. coli, while other strains were more sensitive to Cu(II) complexes than uncoordinated metformin ligands. Overall, **92** demonstrated the strongest inhibitory potential; however, it was not compared with the respective inorganic Cu(II) salt or the clinically used antibiotics [159].

Olar et al. prepared mono- and dinuclear tetracoordinate Cu(II) complexes **94**–**99** (Figure 4) according to the synthetic procedures described earlier [103,104,143,160]. In contrast to structurally similar Ni complexes **73**–**75**, Cu(II) complexes **94**–**96** were not significantly active against *E. faecium*, *P. aeruginosa* and *C. albicans* and completely devoid of activity against *S. aureus*, *E. coli* and *C. parapsilosis* [143]. Only **96** demonstrated stronger inhibitory activity than its Ni analogue against the *E. faecalis* strain. While **93**–**95** did not induce reasonable cytotoxicity in tested cancer cell lines, Cu(II) complexes **94**–**96** induced significant apoptosis and necrosis in HCT8 cell lines, which was associated with their ability to interfere with the cancer cell cycle and cause G2/M phase arrest [143]. Similar to **83** and **84**, **94** and **95** presented good drug-like features, but only **95** displayed reasonable intestinal absorption and suitable blood-brain-barrier (but not central nervous system) permeability [143]. In addition, all tested complexes with macrocyclic ligands strongly inhibited protease activity. Subsequently, compounds **97** and **98** were prepared according to the previously published synthetic procedures [161] and subjected to testing against 82 gram-negative strains of *E. coli*, *K. pneumoniae* and *E. cloacae*, which were isolated from different surfaces in the hospital environment [160]. Dinuclear complex **97** demonstrated significantly higher antibacterial activity than **98**, probably due to the presence of two active metal centers. The most pronounced activity was observed in *E. coli* strains (MIC 18–1250 μg/mL), followed by *K. pneumoniae* and *E. cloacae* (MIC 312.5–1250 μg/L) [160]. In another work, the cytotoxicity of **98** was tested against mouse muscle C2C12 cells and human liver carcinoma HepG2 cells [129]. It was shown that **98** was devoid of toxicity, indicating that it can be safely used as an antibacterial agent or for other purposes. 

Aiming to understand the role of metal center, the antibacterial activity of the complex **99** was compared to the respective inorganic Cu(II) salt and metformin [103]. Both metformin and **99** demonstrated very weak activity against *E. coli*, *S. enteritidis*, *S. aureus* and *C. albicans* (MIC 512–1024 μg/mL), while other strains, namely, *P. aeruginosa*, *B. subtilis* and especially *L. monocytogenes,* were significantly more sensitive to **99** than to metformin (MIC 4–256 μg/mL). However, the corresponding inorganic Cu(II) salt was even more active against all tested strains, indicating the origin of antibacterial activity in **99 [103]**. The counterion exchange from perchlorate to acetate resulted in the formation of complex **100** with a completely different antibacterial profile [104]. Additionally, **100** was only marginally active against *E. coli*, *L. monocytogenes, S. aureus* and *C. albicans* (MIC 512–1024 μg/mL) and moderately active against *S. enteritidis, P. aeruginosa* and *B. subtilis* (MIC 128–256 μg/mL), while the respective Cu salt was mostly devoid of antibacterial activity. Subsequently, the ability of **99, 100** and respective Cu(II) salts to inhibit colonization of the eukaryotic cells by *S. aureus* and *P. aeruginosa* was investigated. It was shown that all compounds completely abolished the colonization of *P. aeruginosa*; however, only **99**, but not a Cu salt, could abolish the colonization of *S. aureus*. These results indicate the potential of **99** to prevent bacterial biofilm formation on hospital-related surfaces and prosthetic devices. 

Structurally similar complexes **99**–**103** were prepared from Cu perchlorate hexahydrate and biguanide ligands, which were in situ generated via the nucleophilic addition of corresponding amines to dicyandiamide [162]. The X-ray diffraction of **99** and **101** confirmed that perchlorate anions were not coordinated to a Cu(II) center but resided in the outer coordination sphere of the complexes. Additionally, **99** and **100** showed considerable antibacterial activity against *E. coli*, *S. typhimurium*, *S. aureus* and *B. cereus* at 1.25 mg/mL concentrations, although they were less effective than standard antibiotics amikacin and gentamicin [162]. Slightly bulkier complex **101** did not show any activity against *E. coli* and *B. cereus* even at 12.5 mg/mL. In addition, the DNA binding properties of all complexes were tested using UV spectroscopy, and it was suggested that all complexes can interact with DNA either via electrostatic or hydrogen bonding interactions [162].

Similar to tetracoordinate complex **98** with two chloride anions in the outer coordination sphere, the hexacoordinate chlorido-complex **104** was moderately active against various bacterial and fungal strains but not active against the *A. flavus* fungal strain [105]. No significant differences were observed between Cu(II) complex **104** and Ni(II) complex **70**, which were studied under identical experimental conditions [105]. In addition, **104** did not show strong antiproliferative effects against MCF-7 and HeLa cancer cell lines (IC_50_ > 50 μM) [163]. Another hexacoordinate complex **105** with monodentate aqua and DAB ligands was moderately active against *E. faecalis*, *K. pneumoniae* and *Shigella* and not active against *S. aureus*, *C. albicans* and *A. niger*. In general, with the exception of *Shigella*, Cu(II) complex **105** was more active than structurally similar Ni(II) and Co(II) complexes **71** and **37** and was similarly active as Zr(II) complex **9**. 

Besides antibacterial properties, DNA binding, antioxidative and antidiabetic properties of Cu(II)-metformin complexes were also investigated. Complex **106** demonstrated some antihyperglycemic activity in rats with STZ-induced diabetes, as well as DNA binding properties, which were comparable with structurally similar Ni(II) complex **70** and Co(II) complex **48** [132]. Hexacoordinate heteroleptic Cu(II) complexes with metformin and amino acid chelating ligands **107**–**109** demonstrated quasi reversible electrochemical behaviour; therefore, the DNA binding properties of **108** and **109** were studied using cyclic voltammetry [164]. Based on the pronounced decrease in peak currents, it was confirmed that **108** and **109** formed DNA-bound Cu(II) complexes at the electrode surface, probably via metformin fragment. It was hypothesized that these redox-active complexes might be involved in the dismutation of superoxide and peroxide radicals. As expected, complexes **107**–**109** demonstrated the ability to inhibit superoxide dismutase and catalase [132]; however, the desirable effects were achieved only at high mM concentrations, which is not desirable for potential anticancer drug candidates. 

Interestingly, several Cu(II) complexes were investigated as potential herbicides for effective weed management [165,166]. The assumption was based on the ability of redox-active Cu(II) complexes to decrease GSH/GSSG ratio leading to the inhibition of protein synthesis and suppression of cell division [62,167]. A series of metformin-derived compounds were prepared by the condensation of 2-thiophene- or 2-imidazolecarboxaldehyde with 2-guanidinobenzimidazole or 2-benzothiazolyl-guanidine [165]. The subsequent coordination to a Cu(II) center achieved by in situ electrochemical method, resulted in the formation of tetracoordinate square planar Cu(II) complexes **103**–**108** with bidentate metformin-derived ligands in 79–92% yields (Figure 4) [165].

The effects of **110**–**115** (Figure 5) and their respective free ligands on the photosynthetic activity of photosystem II (PSII) were studied using photochemically active fragments from spinach leaves [131]. Photochemical changes were quantified based on the PSII chlorophyll fluorescence yield. In general, Cu(II) complexes demonstrated stronger inhibitory effects than the respective ligands. The following major differences were observed with respect to the 5-membered thiazole ring: when NH was replaced with the S atom, the inhibitory activity of Cu(II) complexes increased by more than 10-fold (e.g., 6.3% and 63.4% for **110** and **111**, respectively). No marked differences were observed between **110**, **112** and **114** or between **111**, **114** and **115**. Similar trends were observed with respect to the Cu(II) inhibition of PSII carbonic anhydrase (CA) activity and α-carbonic anhydrase (α-CA) activity in bovine erythrocytes. In particular, a total of 100% inhibition of CA was observed by complexes **113** and **115** (100 μM). While uncoordinated ligands did not induce marked photochemical changes, they demonstrated significant inhibition of CA and α-CA (39.4–78.9% and 5.6–50.9%, respectively). As expected, all complexes inhibited glutathione reductase (GR) from chloroplasts at the nanomolar level, and the highest inhibitory was observed for complex **112** (IC_50_ = 0.025 nM). In order to investigate the mechanism of GR inhibition, the activity of reduced and oxidized forms of GR from *S. cerevisae* in the presence of Cu(II) complexes **110**–**115** and respective ligands was studied in a time-dependent manner. The oxidized form of GR was inhibited by both complexes and ligands, while the reduced form of GR was inhibited only by the complexes, indicating their different mechanisms of action. It was suggested that Cu(II) ions and the ligands might act synergistically, where Cu(II) ions could cause initial oxidation of the enzyme and the ligands subsequently induce irreversible enzyme destruction [166].

In recent years, Au(I) and Au(III) complexes have gained popularity as promising anticancer drug candidates due to their high propensity for intracellular enzymes [168,169,170]. According to Pearson’s theory, Au atoms have a high propensity for “soft” ligands, such as thiols, and therefore, Au complexes readily target thioredoxin reductase (TrxR), GR and other thiol-containing biomolecules that are overexpressed in cancer cells [168]. Besides human thiol-containing enzymes, Au complexes were also reported to target bacterial TrxR and glutathione-dependent enzymes, leading to the efficient inhibition of bacterial respiration [168,171,172,173]. Most of Au-metformin complexes are discussed in the context of their anticancer activity; however, the antimicrobial activity was also reported. Coordination of 3 equiv. of metformin to an Au(III) center yielded a mononuclear octahedral Au(III) complex **116** (Figure 28) [105]. This complex demonstrated moderated inhibition of all tested bacterial and fungal strains (zone of inhibition of 9–20 mm/mg), which was 1.4–4 times lower than the activity of tetracycline. **116** was more than 1.4-times more active against gram-positive *B. subtilis* strains than gram-negative *E. coli* and *P. aeruginosa* strains and more than two times more active than another gram-positive *S. aureus* strain [105].

Chemical attachment of metformin and its analogues to various cyclometalated Au(III) backbones yielded a series of auraformins **117**–**123** (Figure 29) [50,174,175,176]. Initially, Che et al. prepared moderately water-soluble complexes **117** and **118**, which demonstrated anticancer activity in a low micromolar range in a panel of cancer cell lines (IC_50 (72h)_ ≈ 1.5–17.1 μM) and a high degree of selectivity towards cancer cells vs. healthy lung fibroblasts (SF = 2–15) [174]. The cytotoxicity of complexes **117** and **118** against cervical epithelial carcinoma HeLa cells and melanoma B16 cells was comparable to cisplatin; however, **117** and **118** were 5–8 times more active than cisplatin when tested against hepatocellular carcinoma PLC cells and breast carcinoma MDA-MB-231 cells. It was shown that upon interaction with intracellular GSH, **117** and **118** formed GSH adducts, such as [(C^^^N)Au(III)(GSH)_n_], where C^^^N is a cyclometalated backbone and n = 1, 2. These adducts caused extensive cytoplasmic vacuolization and endoplasmic reticulum (ER) stress. In addition, complex **117** caused prominent anti-angiogenic properties at sub-cytotoxic concentrations [174]. Based on these observations, Babak and Ang et al. hypothesized that fine-tuning of the cyclometalated fragment would allow for the prodrug activation and release of biguanide ligands, leading to the complementary action with [(C^^^N)Au(GSH)_n_] fragments in cancer cells [50]. Since Au complexes readily target TrxR, leading to the interference of mitochondrial function, and metformin is a well-known energy disruptor, targeting mitochondrial complex I, we hypothesized that these two components might synergistically disrupt mitochondrial processes in metabolically-dependent cancers, such as triple-negative breast cancers (TNBCs) [50]. Similar to the observations of Che et al., complexes **119**–**123** induced cytotoxicity in MDA-MB-231 cells in a low micromolar range and were at least three times more active than cisplatin and more than 100–1000 times more active than metformin. All complexes induced great selectivity towards cancer cells vs. healthy hepatocytes and cardiomyocytes. In particular, **123** demonstrated prominent anticancer activity (IC_50_ = 0.72 ± 0.08 μM) (Figure 6A). We showed that the anticancer activity of complexes was at least partially dependent on their reactivity towards GSH [50]. The least active complex, **121,** released phenformin ligand without any activation by GSH, indicating its lower stability in aqueous media. In contrast, the most active complex **123** demonstrated time-dependent release of metformin only in the presence of GSH, in agreement with our hypothesis (Figure 6B). Complex **122** released metformin only upon heating and was at least ten times less active than **123**. The lead drug candidate **123** significantly inhibited mitochondrial respiration in TNBC cells and induced ER stress. We showed that induction of integrative stress forced cancer cells to activate various pro-survival responses, such as metabolic reprogramming, UPR and autophagy; however, **123** effectively shut down pro-survival attempts of cancer cells, resulting in the induction of apoptosis. Subsequently, these observations were confirmed by an independent group of researchers [175]. Inspired by the promising in vitro results, we verified the efficacy of **123** in an orthotopic mammary fat pad animal model, which realistically recapitulates the TNBC environment in contrast to commonly used xenograft models (Figure 6C). A marked reduction of tumor burden (Figure 6D) and the formation of large areas of tumor necrosis were caused by **123** [50]. In addition, tumors were characterized by the infiltration of inflammatory cells, suggesting the activation of an immune response. To conclude, complex **123** might be efficient in TNBC patients with a high risk of metastasis and relapse, and it is currently undergoing advanced preclinical investigations [176].

### 2.10. Group XII (Zn, Cd, Hg)

Group 12 consists of Zn, Cd and Hg. While Cd and Hg do not play any physiological role and are highly toxic, the nutritional value of Zn has been known for a very long time [177]. Zn is considered an important chemical element that participates in various biological processes [178]. Zn plays an indispensable role in modulating the function of various enzymes and proteins and acts as an endogenous modulator of neuronal activity [179]. In addition, Zn-based compounds possess a broad range of antimicrobial activity and are commonly used as additives for dental and dermatological purposes [180]. Therefore, several Zn(II) complexes with metformin were prepared with the aim of investigating the role of the Zn metal center in their antimicrobial activity (Figure 7) [103,104,105,124].

The tetrahedral complex **124**, where metformin acts as a monodentate ligand, was tested against *E. coli*, *K. pneumoniae*, *P. aeruginosa*, *B. subtilis* and *S. aureus* in comparison with a structurally similar Co(II) complex **36** and square-planar Pt(II) and Pd(II) complexes **88** and **89** [124]. In agreement with the broad antimicrobial activity spectrum of Zn(II), **124** demonstrated strong inhibitory activity against all strains (MIC 32–128 µg/mL) and was 4–32 times more active than other metal complexes [124]. Similarly, complex **125** with two bidentate metformin ligands demonstrated the highest inhibitory activity against various gram-positive and gram-negative strains, which was comparable to or slightly less active than the activity of tetracycline [105]. It should be noted that the antibacterial activity of **125** was higher than the activity of metformin complexes based on Mn(II), Fe(III), Ni(II), Cu(II), Mg(II), Pt(IV), Au(III) and Pd(II) metal centers. However, this complex was devoid of any activity against the *A. flavus* fungal strain [81]. The inhibitory activity of octahedral complexes **126** and **127** was weaker than tetracoordinate complexes, although the results cannot be directly compared due to the differences in experimental conditions [103,104]. In particular, complex **126** was characterized by marginal activity against all strains in the tested panel (MIC = 128–1024 µg/mL) [104]. When the activity of **126** and **127** was compared with the respective inorganic Zn(II) salts under identical experimental conditions, the inorganic salts demonstrated higher inhibitory potential than the metformin complexes. 

In contrast to Zn, the therapeutic potential of Cd complexes is hindered by the severe health adverse effects associated with Cd exposure. Therefore, the investigation of Cd(II)-metformin complexes might be more interesting from the fundamental point of view. Heteroleptic octahedral Cd(II) complexes **128** and **129** with metformin and DAB or glimepiride ligands were prepared starting from CdCl_2_·H_2_O and their antimicrobial properties were investigated in comparison with uncoordinated ligands or Cd(II) salt (Figure 30) [71,181]. As expected, none of the ligands or Cd(II) salt was active against any bacterial or fungal strain in the panels [71,181]. Conversely, complex **128** demonstrated excellent inhibitory potential against *K. pneumoniae*, *S. aureus* and *Shigella* bacterial strains, which was comparable to moxifloxacin, and similar inhibitory activity against *A. niger* fungal strains as nystatin [71]. It should be noted that structurally similar metformin complexes with Co(II), Ni(II), Cu(II), Zr(IV), Pd(II) metal centers did not show any activity against *A. niger* [71]. Similarly, complex **129** strongly inhibited *E. coli*, *K. pneumoniae*, *P. aeruginosa*, *P. vulgaris* and was particularly active against *S. aureus* [181]. For certain strains, the activity of **129** even exceeded the activity of the antibiotic amikacin. 

### 2.11. The Role of the Metal Center in the Biological Activity and Potential Toxicity of Pre-Formed Metal-Metformin Complexes

Taking into account the well-known role of metformin and other clinically used biguanides in the treatment of diabetes and various infections, as well as the epidemiological evidence linking metfomin and reduced cancer risks, it is not surprising that the majority of metformin-metal complexes were investigated in the context of their antidiabetic, antimicrobial and anticancer properties. In Figure 8 and Table A1, we summarized the lead metal-metformin complexes with the most prominent antibacterial, antifungal, anticancer and antidiabetic activity. 

Since various structurally different V complexes generally exhibit antidiabetic properties [74,75], it was hypothesized that coordination of antidiabetic drug metformin to the V center might result in their synergistic action. The activity of V-metformin complexes was investigated in rats with chemically- or HC diet-induced diabetes. Although V-metformin complexes **10**, **14** and **15** were able to reduce blood glucose levels more efficiently than uncoordinated metformin, no marked improvement in comparison with other V-complexes was observed [76,77,79]. In contrast, when metformin was introduced into the structure of decavanadate [V_10_O_28_]^6−^ as a counterion, the solubility of metforminium decavanadate **16** considerably improved due to the additional hydrogen bonding with the metformnium cation [81]. As a result of more favorable pharmacokinetic properties, metforminium decavanadate **16** was markedly more active than metformin, sodium decavanadate or V salts in various insulin-dependent and insulin-independent animal models [85,86]. Importantly, **16** was also able to improve diabetes- and obesity-associated memory deterioration [87].

Besides V, metformin complexes with other metal centers, such as Nd, Cr, Ni, Cu, Co and Ru, were also investigated in animals with induced diabetes [69,97,98,123,132]; however, only Cr-metformin complexes **20** and **21** demonstrated significantly improved hypoglycemic effects in comparison with metformin and respective Cr salts [97,98]. The observed profound differences in the hypoglycemic activity of various metal-metformin complexes indicate the unambiguous role of V and Cr metal centers in their antidiabetic mechanism of action. It is believed that one of the mechanisms of V-mediated insulin signaling is based on the inhibition of protein tyrosine phosphatase 1b (PTP1B), which is a key enzyme that inactivates insulin receptor [182]; however, the insulin-independent mechanisms were also reported [183]. Cr complexes were also shown to affect insulin receptors, but independently of PTP1B regulation [184]. Importantly, despite improved antidiabetic effects of V- and Cr-metformin complexes, their effects on the metal metabolism should be considered with caution. At higher doses, V- and Cr-metformin complexes might cause unwanted toxicity as a result of the alteration of essential trace element homeostasis. For example, non-insulin dependent diabetic patients were characterized by Cr and V disbalance [185,186]; hence, large doses of Cr- and V-based antidiabetic drugs might exacerbate the already compromised metal status and contribute to the development of insulin resistance.

Based on the analysis of the existing literature, the majority of reported metformin-metal complexes, including Y, La, Ce, Sm, Zr, V, Cr, Mn, Fe, Co, Ir, Ni, Pd, Pt, Cu, Au, Zn and Cd, were tested against various panels of bacterial and fungal strains using standard antibacterial assays, such as the disk agar diffusion method. Since these complexes were tested under different experimental conditions, their antimicrobial efficacy cannot be directly compared. However, most metal complexes demonstrated improved activity in comparison with uncoordinated metformin. In order to estimate whether the antibacterial and antifungal effects of metal complexes solely originate from the metal or rather from additive/synergistic effects of the metal center and biguanide ligands, some of the complexes were compared to the respective inorganic salts. It was shown that several complexes, such as Cr(III) complex **21**, Ni(II) complexes **66** and **67**, Cu(II) complex **100** and Cd(II) complex **128** were indeed more active than uncoordinated ligands and respective inorganic salts, indicating the importance of metformin coordination to the metal centers for enhancement of their antimicrobial properties.

Although the antimicrobial role of various metal surfaces has been recognized since ancient times, the clinical use of metal-based antimicrobial compounds is very limited. Recently, Frei et al. performed the antibacterial screening of 906 structurally different metal complexes with various metal centers [187], which also included some of the Ir(III) complexes with metformin and its derivatives (**50**–**62**) described earlier [137]. Surprisingly, more than 25% of metal complexes were active against bacterial and fungal strains, and 9.7% out of the 906 complexes were active and non-toxic to human cells. This hit-rate is markedly higher than the hit-rate for organic molecules, which does not typically exceed 2%, suggesting that metal complexes hold great potential as antibiotics. It should be noted that the most active and non-toxic metal complexes in the screen were characterized by the presence of Ru, Ag, Pd and Ir centers, while Cd and Pt complexes were active, but expectedly toxic [187]. While most of the tested Ir(III)-biguanide complexes were not active against gram-negative bacterial strains, they demonstrated one of the strongest inhibitory effects against gram-positive bacterial and fungal strains among all tested compounds in the library [187]. Similarly, in comparison with other metformin-metal complexes described in this review, Ir(III) complexes, such as **58** (Figure 8) and Cd(II)-biguanide complexes **128** and **129** demonstrated one of the strongest antimicrobial properties [71,137,181]; however, we do not foresee clinical success of Cd(II)-metformin due to their high toxicity and poor selectivity towards bacterial cells. Complexes with endogenous metals, such as Cu and Zn, were considered as “underperformers” in the large screen of 906 complexes, possibly due to the more accelerated ligand substitution in comparison with the second and third row elements and premature decomposition of the complexes before reaching the desired biomolecular target [187]. On the contrary, Cu(II) and Zn(II) complexes with metformin and other biguanides, such as **99** or **125** (Figure 8), showed marked inhibition of bacterial and fungal growth [71,105]. This could be explained by the high stability of metformin complexes in physiological solutions due to the chelating effect of bidentate biguanide ligands. In addition, La(III)-metformin complex **2**, Pt(IV) complex **91** and Au(III) complex **116** (Figure 8) strongly inhibited the growth of several bacterial and fungal strains [68,105]. The lipophilicity of metal complexes played a determining factor in their antibacterial and antifungal activities, since more lipophilic complexes showed more efficient internalization inside bacterial cells, resulting in stronger inhibitory activity [137]. 

Besides antimicrobial applications, various metal-metformin complexes, including Mn, Ru, Co, Ir, Ni, Pd, Pt, Cu and Au, were tested as potential anticancer agents. Even though the results of cytotoxicity assays cannot be directly compared due to different cancer cell lines and inconsistent drug treatment time, the analysis of literature data revealed that the most significant cytotoxicity was exhibited by Ir(III), Au(III) and Pt(IV) and Ru(II)-metformin complexes [123,138,155]. Ir(III) complexes **63**–**65** were cytotoxic in A549, HeLa and MCF7 cancer cell lines under normoxic and hypoxic conditions (IC_50_ (44 h) ≈ 3–30 μM). In particular, complex **63** (Figure 8) was characterized by strong cytotoxicity in the low micromolar range (IC_50_ (44 h) ≈ 1.4–5.3 μM) [138]. Similarly, Pt(IV) complex **90** (Figure 8) showed submicromolar cytotoxic effects in a cisplatin-sensitive P388 cell line (IC_50_ (48 h) ≈ 0.86 μM) and low micromolar cytotoxic effects in a P388/CDDP cisplatin-resistant cell line (IC_50_ (48 h) ≈ 4.38 μM) [155]. Ru(II) complexes **34** and **35** (Figure 8) exhibited anticancer effects against MDA-MB-231, A549 and A2780 cancer cell lines in the micromolar concentration range (IC_50_ (24 h) ≈ 8–29 μM) and were not toxic against healthy embryonic kidney HEK293 cells [138]. Finally cyclometalated Au(III)-metformin complexes **117**–**123** were tested against a broad panel of cancer cell lines, including HeLa, B16, PLC, MDA-MB-231, SUNE1, U87, A2780 and A2780 cis and showed wide range of cytotoxicity (IC_50_ (72 h) ≈ 0.15–47 μM), as well as certain degree of selectivity towards cancer cell lines [50,174]. In particular, complex **123** (Figure 8) exhibited marked cytotoxicity in a submicromolar concentration range (IC_50_ (72 h) ≈ 0.15–0.72 μM) [50]. In addition, **90** and **123** significantly reduced tumor burden in vivo [50,155].

Interestingly, one of the strongest antibacterial and anticancer activity was exhibited by structurally different biguanide complexes with Ir(III), Au(III), Cu(II), Pt(IV) and Ru(II) centers. It is possible that, despite certain differences between bacteria and cancer cells, the complexes might undergo similar mechanisms of action. It is known that metformin is devoid of strong antibacterial and anticancer in vitro activity due to its inability to efficiently internalize into bacterial or cancer cells. Conversely, the strong antimicrobial activity of redox-active Ir(III), Au(III), Cu(II) and Pt(IV) complexes with metformin and its derivatives might be linked to their ability to selectively deliver biguanides through bacterial or cancer cell walls and subsequently release the active fragments upon reduction or substitution reactions with intracellular thiols or other biomolecules. Subsequently, the uncoordinated metformin might exert its antibacterial action through binding with endogenous metal ions from metalloenzymes or DNA intercalation, while liberated metal fragments might confer synergistic effects through interaction with other metal-binding biomolecules.

## 3. Conclusions and Future Outlook

Since the mechanism of action of metformin and other biguanides is at least partially dependent on intracellular metal binding, administration of pre-formed metal complexes with a pre-determined biguanide/metal ratio might significantly potentiate their on-target biological activity. We demonstrated that coordination of metformin and its analogues to metal centers often results in enhanced intracellular penetration, reduced drug resistance and synergistic action with biologically active metals, which therefore might be a viable strategy to improve the pharmacological activity of biguanides. However, it became apparent that despite extensive research on metformin-metal complexes, most of the studies lacked structure-activity relationships and in-depth investigation of the mechanism of action of metal complexes in the relevant disease models. In addition, the clinical use of metals is often believed to be associated with unwanted toxicity and side-effects; therefore, it is important to study the toxicity profile of novel metal-based drug candidates in order to understand their limitations for future therapeutic applications.

## Data Availability

Data sharing not applicable.

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
