# Peer review of "Biological Properties of Transition Metal Complexes with Metformin and Its Analogues"

_pharmaceuticals, 2022, doi:10.3390/ph15040453_

Round 1

Reviewer 1 Report

The authors summarized the transition metal complexes with metformin. In general, it is quite good for readers. There are several questions the authors needs to be explain.

1, Does the transition metal complexes with metformin have some effects on the metal metabolism? For example, It is reported that 363,447 people reported to have side effects when taking Metformin. Among them, 166 people (0.05%) have Iron deficiency. 

2, Does the transition metal complexes with metformin loss activity of metformin or the function of metal?  What is the role of the complex? For example, Mn is a cofactor of many enzymes. The decreased Mn will affect Arginase activity. 

Author Response

Reviewer 1:

>>The authors summarized the transition metal complexes with metformin. In general, it is quite good for readers. There are several questions the authors needs to be explain.

We thank the reviewer for a positive evaluation of our work. The suggestions are very reasonable and definitely improved the quality of the review. 

>> 1, Does the transition metal complexes with metformin have some effects on the metal metabolism? For example, It is reported that 363,447 people reported to have side effects when taking Metformin. Among them, 166 people (0.05%) have Iron deficiency. And >> 2, Does the transition metal complexes with metformin loss activity of metformin or the function of metal?  What is the role of the complex? For example, Mn is a cofactor of many enzymes. The decreased Mn will affect Arginase activity. 

We have added the new section “The role of the metal center in the biological activity and potential toxicity of pre-formed metal-metformin complexes” (highlighted in yellow), where we explained the role of the metal and metformin moieties, as well as their potential effects on metal metabolism.

Reviewer 2 Report

Comments and suggestions for improving the manuscript:

I suggests you to introduce a graphical representation that summarize all the biological actions investigated for metal complexes of metformin and its analogues.

Please carefully verify the correspondence between the content of the article and of the table 1 and the references mentioned. For exemple, I did not find the article that describe the complex 124, and some inconsistencies.

The expression „on the contrary” appears 16 times in text; alternatively, „contrariwise” could be used sometimes.

Please even color for all structures.

Please correct the conjunction “и” as „and” (rows 290, 292, 293, 672): also „obaserved” as „observed” (row 275)

Row 576, 577: (selectivity factor (SF)  values ranges between 8 and >256), instead of (SF > 8-256).

Scheme 21: 2X- instead of X2; similar observation for compounds 66, 67, 68 in Table 1

In the structure of compound 87, dimethylsulfoxide molecule is S-bonded to Pt(II); a counter anion Cl- is missing (see ref. 118: „L'unité asymétrique de (I) contient un anion Cl- et un cation cis-chloro-S-(dimethyl sulfoxyde)(metformine)- platine(II)”.

Row 715 „..has been tested again(st?)”

Row 778 It is not clear what is the Zr(II) complex 2.

Rows 803, 804 In my opinion, the sentence „The major differences were observed with respect to the 5-membered thiazole ring: when  NH was replaced with the more electronegative S atom...” is wrong. The electronegativities for various N containing groups are around 3, while S atom has the electonegativity 2.5. (https://www.journal.csj.jp/doi/pdf/10.1246/cl.1982.1003). My suggestion is to reformulated the sentence as follows: „The major differences were observed with respect to the 5-membered thiazole ring: the authors assume that when NH was replaced with the more electronegative S atom....”

In the section „Conclusions and future outlook”, more consistent observations must be included; for example, what is/are the most active compound/compounds for each biological activity studied, what are the compounds active on resistent bacteria etc.

Author Response

Reviewer 2:

>>Comments and suggestions for improving the manuscript:

We thank the reviewer for the valuable suggestions. We have addressed all the comments below. 

>>I suggests you to introduce a graphical representation that summarize all the biological actions investigated for metal complexes of metformin and its analogues.

Thank you, the figure has been added.

>>Please carefully verify the correspondence between the content of the article and of the table 1 and the references mentioned. For example, I did not find the article that describe the complex 124, and some inconsistencies.

Thank you, the table has been corrected.

>>The expression „on the contrary” appears 16 times in text; alternatively, „contrariwise” could be used sometimes.

Thank you, indeed. Instead of “on the contrary” we used “in contrast”, “conversely” and “on the other hand”.   

>> Please even color for all structures.

We were not sure whether the reviewer suggested to leave only black colour for all structures or use similar black and blue colouring for all figures. We changed the colour of metformin moieties in all complexes to blue as suggested by Reviewer 3. Please advise if this change is not satisfactory.  

>> Please correct the conjunction “и” as „and” (rows 290, 292, 293, 672): also „obaserved” as „observed” (row 275)

Thank you, corrected.

>> Row 576, 577: (selectivity factor (SF) values ranges between 8 and >256), instead of (SF > 8-256).

Thank you, corrected.

>> Scheme 21: 2X- instead of X2; similar observation for compounds 66, 67, 68 in Table 1

Thank you, corrected both in the text and the table.

>> In the structure of compound 87, dimethylsulfoxide molecule is S-bonded to Pt(II); a counter anion Cl- is missing (see ref. 118: „L'unité asymétrique de (I) contient un anion Cl- et un cation cis-chloro-S-(dimethyl sulfoxyde)(metformine)- platine(II)”.

Thank you, corrected.

>> Row 715 „..has been tested again(st?)”

Thank you, corrected.

>> Row 778 It is not clear what is the Zr(II) complex 2.

Thank you, corrected.

>>Rows 803, 804 In my opinion, the sentence „The major differences were observed with respect to the 5-membered thiazole ring: when  NH was replaced with the more electronegative S atom...” is wrong. The electronegativities for various N containing groups are around 3, while S atom has the electonegativity 2.5. (https://www.journal.csj.jp/doi/pdf/10.1246/cl.1982.1003). My suggestion is to reformulated the sentence as follows: „The major differences were observed with respect to the 5-membered thiazole ring: the authors assume that when NH was replaced with the more electronegative S atom....”

Thank you, corrected.

>>In the section „Conclusions and future outlook”, more consistent observations must be included; for example, what is/are the most active compound/compounds for each biological activity studied, what are the compounds active on resistent bacteria etc.

We have added the new section “The role of the metal center in the biological activity and potential toxicity of pre-formed metal-metformin complexes”, where we explained the role of the metal and metformin moieties, as well as their potential effects on metal metabolism and toxicity.

Reviewer 3 Report

The manuscript entitled „Biological properties of transition metal complexes with metformin and its analogues“ was submitted by D. A. Rusanov et al. to the journal “Pharmaceuticals” in order to be considered for publication as a review. The manuscript provides a comprehensive overview of metal complexes that bear metformin-derived ligands, while focusing on their biological activity. The presentation of the complexes refers to the groups in the periodic table – making the review structured. The submitted manuscript is generally suitable to be published as a review in the journal “Pharmaceuticals”. It is of special interest to scientists working in the field of bioinorganic medicinal chemistry. However, there are some aspects I wish to be addressed by the authors – mainly formal aspects and the consideration of research work in the field of metal complexes serving as bioactive compounds. The embedding and connection of recent/previous reviews in this field would strengthen the existence of the current manuscript.

The authors are kindly requested to satisfy the guidelines to authors regarding the style of presentation of the references in the running text of the manuscript, namely before the punctuation – confer author instructions of MDPI: “In the text, reference numbers should be placed in square brackets [ ] and placed before the punctuation”.

Concerning the structural representation in Figure 1: Maybe the authors can think about considering another tautomeric way of presentation the guanidine, especially the diguanidines, to make the guanidine more conspicuous when looking at the chemical formulae.

A consistent use of case shift with spelling the INN of the compounds should be considered throughout the manuscript. Also with the names of bacteria.

Concerning Figure 2: Why do the authors not consider “Polyhexanide (PHMB)” as antiseptic? Indeed, metformin sulfenamide is a prodrug but it is suggested to term is “antidiabetic”. For example, proguanil is also an prodrug (reduction to form cycloguanil) but is presented as antimalarial compound in the manuscript.

“In light of the COVID-19 pandemic, moroxydine, metformin and other biguanides are considered for the treatment and management of SARS-CoV-2” The authors are asked to consider a recent review: 10.1016/j.tim.2021.03.004

“However, it is not clear whether the observed effects were caused by Cu(II) alone or the combination of Cu(II) and metformin” The authors are asked to add a short comment, a clause describing the general mode of Cu complexes to cause cytotoxicity (cleavage of DNA), e.g., considering work by the group of N. Kulak for example.

The numbering of the paragraphs contains errors. Please revise! Also please revise the use of spaces in the manscript (e.g., with units).

anti-emetic… without hyphen to be consistent?

Scheme 2, structure of compound 6. Why to arrows with chloride? How about n=?

Lines 291, 292, 293: are these signs common? Or is it just me not being familiar with them?

Scheme 6: The phenyl moiety in the structure of compounds 14/15 is distorted. Please adjust!

Lines 673/692: -fold vs. spelling elsewhere in the manuscript. Please unify the style of spelling!

“Subsequently, the antibacterial activity of both complexes and metformin has been tested…” The authors are asked to add a short comment, a clause describing the general mode of Fe complexes to cause antibacterial effects (ferroptosis), e.g., considering work by the group of R. Gust for example.

“Ru(II) and Ru(III) complexes with biological properties gained considerable popularity in the recent decades.” The authors are asked to cite reviews by I. Kostova: 10.2174/092986706776360941 and by S. Y. Lee:  10.2147/DDDT.S275007 for example.

In some Schemes, the ligand L or L’ should be introduced.

Figure 3: either 2X- and X=Cl/OAc or 2X and X=Cl-/OAc- also Scheme 29 for example. Please check all Figures/Schemes regarding this issue.

“Since the discovery of cisplatin, Pt(II) complexes have been extensively investigated for their anticancer properties” .” The authors are asked to cite reviews on that topic, such as by I. Kostova: 10.2174/157489206775246458 or by D. Gibson: https://doi.org/10.1016/j.jinorgbio.2020.111353 for example.

Lines 747-748: I guess citing reference [100] once should be fine.

Figure 5: I am wondering about the blue color. This is reasonable. But why was it omitted in all other structures? Please be consistent throughout the manuscript.

“In recent years, Au(I) and Au(III) complexes have gained popularity as promising anticancer drug candidates” The authors are asked to cite more recent work, such as: https://doi.org/10.3389/fchem.2020.00543 from 2020 and 10.3390/molecules23061410 from 2018.

“Besides human thiol-containing enzymes, Au complexes were also reported to target bacterial TrxR…” It is recommended to cite recent work on that topic by the group of I. Ott for example.

Figure 6: The caption of the figure has no explanation regarding (A), (B), (C), (D).

Scheme 30: The brackets should cover the whole chemical structure.

Author Response

Reviewer 3:

>>The manuscript entitled „Biological properties of transition metal complexes with metformin and its analogues“ was submitted by D. A. Rusanov et al. to the journal “Pharmaceuticals” in order to be considered for publication as a review. The manuscript provides a comprehensive overview of metal complexes that bear metformin-derived ligands, while focusing on their biological activity. The presentation of the complexes refers to the groups in the periodic table – making the review structured. The submitted manuscript is generally suitable to be published as a review in the journal “Pharmaceuticals”. It is of special interest to scientists working in the field of bioinorganic medicinal chemistry.

We thank the reviewer for a positive evaluation of our work. This review is indeed specifically tailored for bioinorganic medicinal chemists and we appreciate very valuable comments of the reviewer, in particular with regards to the connection with the reviews in the field.

>> However, there are some aspects I wish to be addressed by the authors – mainly formal aspects and the consideration of research work in the field of metal complexes serving as bioactive compounds. The embedding and connection of recent/previous reviews in this field would strengthen the existence of the current manuscript.

Following the suggestion of the reviewer, we have added the references for reviews about biological properties of metal complexes for each section. Additionally, we added short comments about the general biological role of each metal in the corresponding sections.

>>The authors are kindly requested to satisfy the guidelines to authors regarding the style of presentation of the references in the running text of the manuscript, namely before the punctuation – confer author instructions of MDPI: “In the text, reference numbers should be placed in square brackets [ ] and placed before the punctuation”.

Thank you, corrected everywhere.

>>Concerning the structural representation in Figure 1: Maybe the authors can think about considering another tautomeric way of presentation the guanidine, especially the diguanidines, to make the guanidine more conspicuous when looking at the chemical formulae.

Noted with thanks. We chose this representation of this tautomeric form based on the recent review of Bharatam et al. Eur. J. Med. Chem. 2021, 219, 113378, where the authours specifically pinpointed the correct and incorrect way of guanidine representation. We have checked bond distances using published crystal structures of metformin and its analogues and agreed with the suggestion of Bharatam et al. 

>>A consistent use of case shift with spelling the INN of the compounds should be considered throughout the manuscript. Also with the names of bacteria.

Thank you, corrected. With regards to the names of bacteria, the names were normalized using the following format; S. aureus, B. subtilis, E. coli, P. aeruginosa except for Shigella, since the authors did not specify the particular type of Shigella bacteria.

>>Concerning Figure 2: Why do the authors not consider “Polyhexanide (PHMB)” as antiseptic?

Thank you, we added polyhexanide to the figure and the text.

>> Indeed, metformin sulfenamide is a prodrug but it is suggested to term is “antidiabetic”. For example, proguanil is also an prodrug (reduction to form cycloguanil) but is presented as antimalarial compound in the manuscript.

Noted with thanks. We added proguanil to the prodrug section. We would like to leave this section since this concept can be used for various diseases. We added a short statement into the relevant section.

>>“In light of the COVID-19 pandemic, moroxydine, metformin and other biguanides are considered for the treatment and management of SARS-CoV-2” The authors are asked to consider a recent review: 10.1016/j.tim.2021.03.004

Thank you, the review was added.

>>“However, it is not clear whether the observed effects were caused by Cu(II) alone or the combination of Cu(II) and metformin” The authors are asked to add a short comment, a clause describing the general mode of Cu complexes to cause cytotoxicity (cleavage of DNA), e.g., considering work by the group of N. Kulak for example.

Thank you, added.

>> The numbering of the paragraphs contains errors. Please revise! Also please revise the use of spaces in the manscript (e.g., with units).

Thank you, corrected. We will also double check during the proofreading stage.  

>> anti-emetic… without hyphen to be consistent?

Thank you, corrected.

>> Scheme 2, structure of compound 6. Why to arrows with chloride? How about n=?

Thank you, the structure was corrected.

>> Lines 291, 292, 293: are these signs common? Or is it just me not being familiar with them?

We assumed that the reviewer is asking about the protein tyrosine phosphatases. We added the full name of each PTP.

>>Scheme 6: The phenyl moiety in the structure of compounds 14/15 is distorted. Please adjust!

Thank you, corrected

>>Lines 673/692: -fold vs. spelling elsewhere in the manuscript. Please unify the style of spelling!

Thank you, corrected

“Subsequently, the antibacterial activity of both complexes and metformin has been tested…” The authors are asked to add a short comment, a clause describing the general mode of Fe complexes to cause antibacterial effects (ferroptosis), e.g., considering work by the group of R. Gust for example.

Thank you, added

>>“Ru(II) and Ru(III) complexes with biological properties gained considerable popularity in the recent decades.” The authors are asked to cite reviews by I. Kostova: 10.2174/092986706776360941 and by S. Y. Lee:  10.2147/DDDT.S275007 for example.

Thank you, added.

>>In some Schemes, the ligand L or L’ should be introduced.

Noted with thanks. We prefer not to overcrowd the synthetic scheme with bulky ligands other than metformin. The reader can easily understand the structure of L or L’ from the complex structure.

>> Figure 3: either 2X- and X=Cl/OAc or 2X and X=Cl-/OAc- also Scheme 29 for example. Please check all Figures/Schemes regarding this issue.

Thank you, corrected everywhere.

>>“Since the discovery of cisplatin, Pt(II) complexes have been extensively investigated for their anticancer properties” .” The authors are asked to cite reviews on that topic, such as by I. Kostova: 10.2174/157489206775246458 or by D. Gibson: https://doi.org/10.1016/j.jinorgbio.2020.111353 for example.

Thank you, added.

>> Lines 747-748: I guess citing reference [100] once should be fine.

Thank you, corrected.

>> Figure 5: I am wondering about the blue color. This is reasonable. But why was it omitted in all other structures? Please be consistent throughout the manuscript.

Thank you, corrected everywhere.

>> “In recent years, Au(I) and Au(III) complexes have gained popularity as promising anticancer drug candidates” The authors are asked to cite more recent work, such as: https://doi.org/10.3389/fchem.2020.00543 from 2020 and 10.3390/molecules23061410 from 2018.

Thank you, added.

>>“Besides human thiol-containing enzymes, Au complexes were also reported to target bacterial TrxR…” It is recommended to cite recent work on that topic by the group of I. Ott for example.

Great suggestion, thank you. We missed that paper.

>> Figure 6: The caption of the figure has no explanation regarding (A), (B), (C), (D).

Thank you, added.

>> Scheme 30: The brackets should cover the whole chemical structure.

Thank you, corrected.

Round 2

Reviewer 1 Report

It is a useful paper for reader, and needs to be published.

Reviewer 2 Report

The authors have made the requested modifications. The manuscript is improved and suitable for publication.

Reviewer 3 Report

The revised version of manuscript entitled “Biological properties of transition metal complexes with metformin and its analogues” by D. Rusanov et al. was submitted in order to be considered for publication as “Review” in the journal “Pharmaceuticals”.

The authors provided a revision that acted on every concern and suggestion as mentioned before. They corrected formal errors, did a diligent job to embed their current study into the landscape of other reviews on bioactive metal complexes.

It is especially praiseworthy that the authors added the section 2.11 to the manuscript – strengthening the impact of their review. Regarding this paragraph, however, I wish to make a very short comment. The authors mention “importance of metformin coordination to the metal centers for enhancement of their antimicrobial properties”. This is often related to the so-called Tweedy’s chelation theory in literature. Later on in the running text, the authors even provide the explanation of this theory in their manuscript: “The lipophilicity of metal complexes played a determining factor in their antibacterial and antifungal activity, since more lipophilic complexes showed more efficient internalization inside bacterial cells, resulting in stronger inhibitory activity.”

Very minor typos, such as @ line 475: Schiff (not: Shiff) (confer lines 319, 321, 432, 531, 534, 535, 538, 539, 747) or @ line 1020: cell lines (not: cell ines), should be corrected during thorough proofreading.

All in all, the authors present a comprehensive review. I support further processing in order to be published. All the best!